# Expandable, Compressible, Mineable: Open-World Thermal Image Restoration

Pu Li [1]    Huafeng Li [* 1]    Yafei Zhang [1]    Wen Wang [2]    Neng Dong [1]    Jie Wen [3]

## Abstract

In open-world settings, thermal infrared (TIR) image degradations continuously emerge and evolve, while most existing all-in-one restoration methods are built on a closed-set assumption and struggle to continually adapt to novel degradations. To address this, we propose ECMRNet, an Expandable, Compressible, and Mineable Restoration Network for open-world TIR restoration from a continual learning perspective. Conceptually, ECMRNet unifies continual degradation learning as an "expand–compress–mine" closed-loop process, enabling sustained adaptation to new degradations with controllable evolution. Structurally, ECMRNet decomposes intermediate representations into group-isolated subspaces, and achieves strict parameter isolation and fast adaptation to new degradations by freezing historical groups and isomorphically expanding new ones. To curb model growth as tasks accumulate, we present Structural Entropy Pruning, which identifies and removes redundant channel groups via two-dimensional structural entropy minimization, achieving information contribution–driven adaptive compression. Moreover, we design a Sub-degradation Knowledge Mining Module that dynamically retrieves and recombines transferable components from historical representations to improve restoration under compound degradations. Experimental results demonstrate that ECMRNet achieves superior overall performance across diverse single and compound degradations while using fewer parameters and lower computational cost. The source code is available at https://github.com/Kust-lp/ECMRNet.

[1]Faculty of Information Engineering and Automation, Kunming University of Science and Technology, Yunnan, China [2]School of Mathematics and Statistics, Yunnan University, Yunnan, China [3]Shenzhen Key Laboratory of Visual Object Detection and Recognition, Harbin Institute of Technology, Shenzhen, China. Correspondence to: Huafeng Li <hfchina99@163.com>.

*Proceedings of the 43rd International Conference on Machine Learning*, Seoul, South Korea. PMLR 306, 2026. Copyright 2026 by the author(s).

## 1. Introduction

In open-world thermal infrared (TIR) imaging, degradations are diverse and continually evolving due to sensor characteristics, ambient temperature variations, and imaging distance (Liu et al., 2025c). Existing all-in-one TIR restoration methods (Pang et al., 2023; Liu et al., 2025c;a) use a shared-parameter model that performs well under a closed set of known degradations, but struggles when the distribution drifts and new degradations emerge: updating the shared backbone induces interference and gradient conflicts, leading to catastrophic forgetting, while full retraining is computationally and operationally costly. As a result, the conventional all-in-one paradigm lacks structural support for continual adaptation, limiting scalability and stability for continual TIR restoration.

Continual learning (CL) provides a principled framework for long-term model evolution, yet remains underexplored for TIR restoration. Existing CL methods mainly include: (i) regularization-based approaches (Aljundi et al., 2018; Zenke et al., 2017; Kirkpatrick et al., 2017; Li & Hoiem, 2017) that protect important past parameters while often fail under strong task conflicts and parameter interference in TIR restoration; (ii) expansion-based methods (Liu et al., 2020; Rusu et al., 2016) that freeze old parameters and add branches to avoid interference, while incur near-linear growth in model size and training cost; and (iii) isolation-based methods (Mallya et al., 2018; Mallya & Lazebnik, 2018) that reserve capacity by freezing part of the network, yet are constrained by a fixed capacity budget for an open-ended stream of new degradations. **Therefore, a core challenge in continual TIR restoration is to enable interference-free continual adaptation to new degradations while preventing unbounded model growth.**

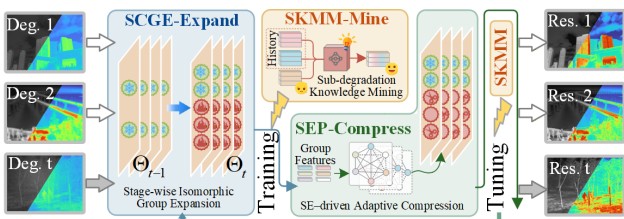

*Figure 1.* Conceptual illustration of the proposed ECMRNet, embodying the "expand–compress–mine" closed-loop mechanism for continual TIR restoration.

To this end, we propose **ECMRNet**, an Expandable, Compressible, and Mineable Restoration Network for open-world TIR degradation from a CL perspective. ECMRNet adopts group-isolated subspaces as the carrier for CL. Built upon a U-Net backbone, it uses group convolutions in each stage to explicitly partition intermediate representations into mutually isolated channel groups, while employing standard convolutions only at the network head and tail for inter-group interaction, retaining representation capacity while enabling high structural expandability. Exploiting the non-interaction across groups, we introduce **Stage-wise Channel Group Expansion (SCGE)**: upon a new degradation, we freeze historical groups and append isomorphic groups at each stage to create new trainable subspaces, enabling strict gradient isolation and fast adaptation without harming prior knowledge. To prevent unbounded model growth as tasks accumulate, we further propose **Structural Entropy Pruning (SEP)**. SEP adopts two-dimensional structural entropy (2D-SE) minimization (Li & Pan, 2016; Xian et al., 2025) as a unified criterion to assess group-level contribution, prunes redundant groups across stages, and then performs lightweight fine-tuning, yielding information contribution-driven adaptive compression. Additionally, we design a **Sub-degradation Knowledge Mining Module (SKMM)** that treats historical sub-degradation representations as a retrievable knowledge base, and dynamically mines and recombines transferable components via sample-adaptive low-rank channel mixing, thereby enhancing restoration under compound degradations. With this "isomorphic expansion–SE compression–knowledge mining" closed loop (Figure 1), ECMRNet enables interference-free continual adaptation while keeping parameter growth under control and effectively reusing historical knowledge to improve compound-degradation restoration. Extensive experiments on multiple TIR datasets show that ECMRNet achieves overall superior performance on three single degradations and four compound degradations, with significantly fewer parameters and lower computational complexity, demonstrating its effectiveness and scalability for open-world TIR restoration. The main contributions of this work are summarized as follows:

- We present **ECMRNet** for open-world TIR restoration, which formulates continual degradation learning as an evolving closed loop of "expand–compress–mine," enabling continual adaptation to new degradations with controllable parameter evolution. To the best of our knowledge, this is the first systematic study addressing open-world TIR restoration.

- We introduce **SCGE** based on group-isolated subspaces, which freezes historical groups and appends new isomorphic groups to achieve strict parameter isolation and rapid adaptation to emerging degradations.

- We propose **SEP**, which performs group-level contribution evaluation and pruning by minimizing two-dimensional structural entropy, enabling information-contribution–driven adaptive compression and effectively suppressing parameter inflation.

- We design **SKMM**, which mines transferable components from historical sub-degradation representations via sample-adaptive low-rank channel retrieval and recombination, substantially improving restoration under compound degradations.

## 2. Related Work

### 2.1. All-in-One Visible Image Restoration.

All-in-one restoration for visible images aims to address diverse degradations with a single model, reducing the cost of training and maintaining task-specific networks. From a technical perspective, mainstream methods can be broadly grouped into three lines: vision large model (VLM)-based, prompt-based, and diffusion-based restoration. **VLM-based methods** leverage semantic priors from VLMs to guide restoration. DA-CLIP (Luo et al., 2024) attaches a trainable image controller to frozen CLIP (Radford et al., 2021) to predict clean-aligned embeddings, alleviating representation mismatch under severe degradations, while VL-UR (Liu et al., 2025b) injects input-aligned CLIP semantics as conditions to improve generalization in complex cases. **Prompt-based methods** encode degradation cues with learnable prompts to modulate intermediate features. PromptIR (Potlapalli et al., 2023) uses prompts for controllable and robust mixed-degradation restoration, and InstructIR (Conde et al., 2024) further replaces handcrafted prompts with free-form language instructions for flexible control. **Diffusion-based methods** exploit clean-image diffusion priors for unified multi-degradation modeling. MPerceiver (Ai et al., 2024) converts CLIP image features into diffusion conditions and uses multi-scale latent prompts from Stable Diffusion (Rombach et al., 2022), whereas DiffUIR (Zheng et al., 2024) aligns shared degradation statistics via selective hourglass mapping to improve conditional diffusion restoration. Beyond these, AdaIR (Cui et al., 2025) modulates features with frequency-aware degradation patterns, and MoCE-IR (Zamfir et al., 2025) routes inputs to experts of different complexity/receptive fields for efficiency.

Recent efforts also extend to open-set or evolving settings: TAO (Gou et al., 2024) performs test-time alignment to unknown degradations via a lightweight adapter with a pre-trained diffusion prior; LIRA (Liu et al., 2020) reduces forgetting via expert growing; and ILAWR (Lu et al., 2025) adopts continual distillation for incremental adaptation in adverse weather removal. However, TIR differ fundamentally from visible images in imaging mechanisms and degrada-

tion patterns (Bao et al., 2024; Harris & Chiang, 1999), making visible-oriented designs hard to directly transfer for effective TIR restoration.

## 2.2. Thermal Infrared Image Restoration.

Early thermal infrared (TIR) restoration mainly targets single degradations—such as contrast enhancement (Wang et al., 2025; Qiu et al., 2024), deblurring (Yi et al., 2023; Zhou et al., 2023), and denoising (Cai et al., 2024; Jiang et al., 2025)—often relying on degradation-specific priors and tailored architectures. Recently, several all-in-one TIR enhancement frameworks have been proposed to handle multiple degradations. TSIRIE (Pang et al., 2023) employs a dual-stream fully convolutional architecture that combines a detail enhancement subnet and a global content preservation subnet. DEAL (Liu et al., 2025c) alleviates the scarcity of high-quality TIR data via dynamic degradation simulation and adversarial training. PPFN (Liu et al., 2025a) incorporates degradation context to guide restoration and adopts progressive training to improve adaptation to compound degradations. SEGD (Li et al., 2026a) models different degradations in a divide-and-conquer manner, estimates degradation type and intensity with the evidential network, and aggregates representations from multiple restoration paths under a SE criterion to improve compound degradation restoration.

Despite these advances, most all-in-one TIR methods assume a fixed, predefined degradation types. Under distribution shift with emerging degradations, updating shared backbones induces interference and catastrophic forgetting, while full retraining is often impractical. This inherent tension suggests that conventional all-in-one paradigms lack structural adaptation mechanisms for open-world TIR restoration, fundamentally limiting both scalability and stability. In contrast, the proposed ECMRNet formulates TIR restoration as open-world continual degradation learning: it supports fast, interference-free degradation adaptation via structured incremental capacity and parameter isolation, achieves controllable model evolution through SE-guided structural pruning, and enhances compound TIR restoration by selectively reusing transferable historical components.

## 3. Method

**In this paper, open-world TIR image restoration can be formally defined as a continual degradation restoration problem.** Let the set of degradations already learned by the model be denoted as $\mathcal{D} = \{D_1, D_2, \dots\}$. A newly arriving degradation is denoted by $D_{new} = \{X^{new}, Y^{new}\}$, where $X^{new}$ and $Y^{new}$ represent the degraded images and reference images, respectively. The model is trained only on the current task $D_{new}$, i.e., $f(\cdot|\Theta_{new}) : X^{new} \rightarrow Y^{new}$. The updated model needs to adapt to $D_{new}$, and preserve

its restoration capability on all previously tasks.

As shown in Figure 2, ECMRNet models continual degradation learning for TIR restoration as an evolving closed loop of dynamic expansion, SE pruning, and knowledge mining: (1) **SCGE** appends isomorphic channel groups within each stage while freezing historical groups, creating new trainable subspaces and enabling strict parameter isolation for fast adaptation to emerging degradations. (2) **SEP** uses 2D-SE minimization as a unified criterion to measure group contributions and prune redundant groups, thereby bounding the growth induced by continual expansion and yielding a controllable, interpretable structural evolution. (3) **SKMM** treats historical sub-degradation group representations as retrievable knowledge and mines/recombines transferable components via sample-adaptive low-rank channel mixing, improving restoration and generalization under compound degradations. Together, these modules turn TIR restoration from static parameter accumulation into structurally constrained continual evolution, enabling joint control of performance and model complexity.

### 3.1. Stage-wise Channel Group Expansion

In continual degradation learning, the key challenge is to add sufficient yet controllable capacity for emerging degradations without disrupting previously learned abilities. We thus propose **Stage-wise Channel Group Expansion** (SCGE), which confines capacity growth to the stage level and expands the network via channel groups, balancing expressiveness, stability, and compressibility. SCGE is built on the group-isolated design of ECMRNet. ECMRNet adopts a U-Net backbone (Figure 2(A)): each stage applies standard convolutions only at the entrance and exit for limited inter-group mixing, while the stage body stacks group residual blocks (GResBlocks). Each GResBlock consists of GNorm, group convolution, GELU, and group channel attention (GCA). This "mix-at-ends, separate-in-middle" design partitions stage-wise features into mutually isolated group subspaces. Since group convolutions have no cross-group computation path, which naturally provides a reliable carrier for parameter isolation in continual degradation learning.

Before task-incremental learning, we perform clean-to-clean self-reconstruction pretraining on clean samples $I^c$ to learn degradation-agnostic representations:

$$\hat{I}^c = f(I^c; \Theta^{ori}), \qquad (1)$$

where $f(\cdot)$ is the backbone, $\Theta^{ori}$ are pretrained parameters, and $\hat{I}^c$ is the reconstruction. After pretraining, the feature extractor, bottleneck, and reconstruction module are frozen for all subsequent tasks, stabilizing the base representation space and output mapping.

When the $t$-th degradation task arrives, SCGE appends new channel groups isomorphically within each stage. This en-

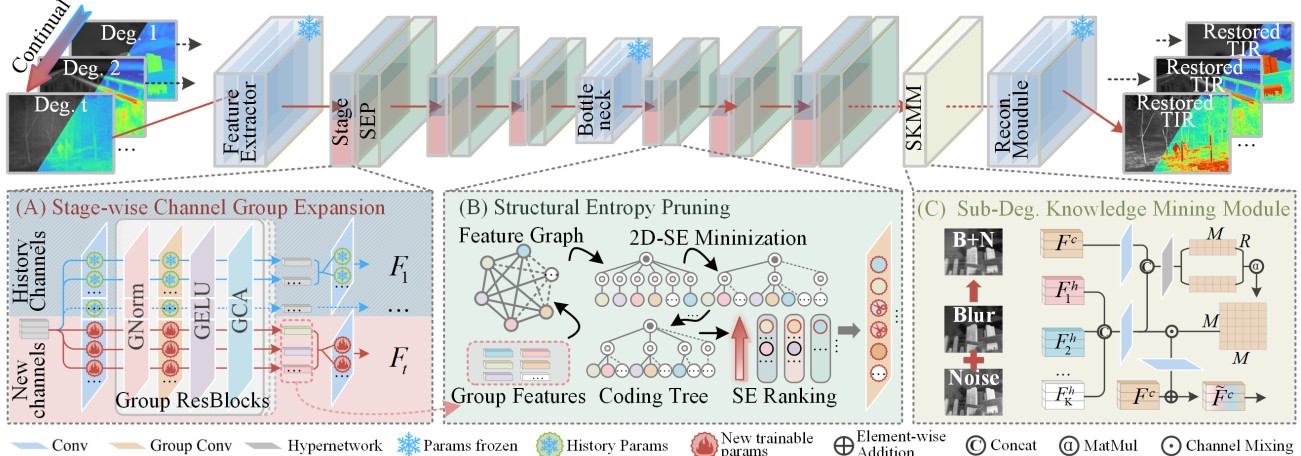

*Figure 2.* Architecture of the proposed ECMRNet. ECMRNet unifies continual TIR degradation learning into an expand–compress–mine closed loop: SCGE expands stage-wise channel groups for interference-free adaptation, SEP prunes redundant groups via 2D-SE minimization to bound model growth, and SKMM dynamically mines transferable components from historical sub-degradation representations to enhance compound-degradation restoration.

sures that each new degradation gains independent capacity at multiple scales. Let $G_s^{(t)}$ and $C_s^{(t)}$ denote the number of groups and channel width of stage $s$ under task $t$, and let each group have width $m_s$:

$$G_s^{(t)} = G_s^{(t-1)} + G_s^{ori}, \quad C_s^{(t)} = C_s^{(t-1)} + G_s^{ori} \cdot m_s, \quad (2)$$

where $G_s^{ori}$ is the group number of the pretrained backbone at stage $s$. At the parameter level, stage-$s$ parameters at time $t$ are decomposed as

$$\Theta_s^{(t)} = \underbrace{\Theta_{s,\text{his}}^{(t-1)}}_{\text{frozen}} \oplus \underbrace{\Delta\Theta_s^{(t)}}_{\text{trainable}}, \quad \Delta\Theta_s^{(t)} \leftarrow \Theta_s^{ori}, \quad (3)$$

where $\oplus$ concatenates along channel groups. The newly added groups $\Delta\Theta_s^{(t)}$ are initialized from the corresponding pretrained parameters $\Theta_s^{ori}$. Due to the absence of cross-group paths in group convolutions, freezing historical parameters $\Theta_{s,\text{his}}^{(t-1)}$ is sufficient for strict gradient isolation, preventing interference and catastrophic forgetting. Moreover, we expand stage-head convolution and add a task-specific stage-tail convolution for the added groups. This keeps historical information paths unchanged while learning new paths independently. Overall, SCGE yields structured, group-wise growth, providing a consistent granularity for subsequent 2D-SE redundancy assessment and pruning.

### 3.2. Structural Entropy Pruning

While SCGE provides strictly isolated subspaces for new degradations, it inevitably increases model capacity as tasks accumulate. Moreover, under the group-isolated design, newly added groups may exhibit weakened competition and converge to functionally overlapping representations. Hence, continual TIR restoration requires an adaptive compression mechanism aligned with the group-wise expansion

granularity to prevent unbounded growth during long-term evolution. Thus, we propose **Structural Entropy Pruning** (SEP), performed after training each degradation task. SEP uses a sample pool to robustly estimate the information contribution of newly added groups, and prunes redundant groups stage-wise to keep model growth and performance jointly controllable. As shown in Figure 2(B), SEP measures redundancy from an information-theoretic view: if a group can be substituted by others, removing it will not significantly increase the optimal coding cost; otherwise, it should be preserved. This criterion naturally matches ECMRNet's group-wise expansion unit, forming an "expand–compress" loop at the same granularity.

Concretely, for each sample $n$ in the pool, we extract stage-wise GResBlocks' outputs and build an intra-group similarity graph $\mathcal{G}_{s,o}^n = (\mathcal{V}_{s,o}^n, \mathcal{E}_{s,o}^n)$ for each newly added group $o \in \{1, \ldots, G_s^{ori}\}$ in the current stage $s$. Vertices $\mathcal{V}_{s,o}^n$ correspond to channel features $\{\mathbf{u}_{s,o,i}^n\}_{i=1}^{m_s}$, and edges are non-negative cosine similarities:

$$e_{ij} = \max\big(\text{Cos}(\mathbf{u}_{s,o,i}^n, \mathbf{u}_{s,o,j}^n), 0\big), \ i \neq j, \ i,j \in \{1, \ldots, m_s\}. \quad (4)$$

On $\mathcal{G}_{s,o}^n$, we compute one-dimensional structural entropy (1D-SE) (Li & Pan, 2016) to quantify channel importance and aggregate channels into a group-level representation:

$$\mathcal{H}_{s,o,i}^{(1)} = -\frac{d_{s,o,i}^n}{v_{\mathcal{G}_{s,o}^n}} \log \frac{d_{s,o,i}^n}{v_{\mathcal{G}_{s,o}^n}}, \quad \mathbf{w}_{s,o}^n = \text{softmax}([\mathcal{H}_{s,o,i}^{(1)}]_{i=1}^{m_s}),$$

$$\mathbf{F}_{s,o}^n = \sum_{i=1}^{m_s} \mathbf{w}_{s,o,i}^n \mathbf{u}_{s,o,i}^n. \quad (5)$$

where $d_{s,o,i}^n$ is the vertex degree and $v_{\mathcal{G}_{s,o}^n}$ is the graph volume (the sum of vertex degrees). $\text{softmax}(\cdot)$ is the Softmax function, $\mathbf{w}_{s,o}^n$ is the aggregation weight, and $\mathbf{F}_{s,o}^n$ is the aggregated group-level feature. This yields a com-

pact SE-driven group feature, enabling subsequent pruning at the group granularity. Given the group-level feature set $\{\mathbf{F}_{s,o}^n\}_{o=1}^{G_s^{ori}}$, we construct an inter-group similarity graph $\mathcal{G}_s^n = (\mathcal{V}_s^n, \mathcal{E}_s^n)$ and find an optimal partition $\mathcal{P}_s^n = \{\mathcal{C}_1, \ldots, \mathcal{C}_k\}$ by minimizing two-dimensional structural entropy (2D-SE):

$$\mathcal{H}^{(2)}(\mathcal{P}_s^n) = -\sum_{\mathcal{C} \in \mathcal{P}_s^n} \left( \frac{g_\mathcal{C}}{v_{\mathcal{G}_s^n}} \log \frac{v_\mathcal{C}}{v_{\mathcal{G}_s^n}} + \sum_{o' \in \mathcal{C}} \frac{d_{o'}}{v_{\mathcal{G}_s^n}} \log \frac{d_{o'}}{v_\mathcal{C}} \right), \tag{6}$$

where $g_\mathcal{C}$ is the cut weight between cluster $\mathcal{C}$ and the rest, and $v_\mathcal{C}$ is the cluster volume. We adopt the near-linear-time CoDeSEG algorithm (Xian et al., 2025) to obtain a 2D-SE-minimized coding tree and its corresponding partition $\mathcal{P}_s^n$, producing clusters with higher within-cluster similarity and stronger cross-cluster separability. This characterizes the redundancy structure among channel groups in an information-theoretic sense. Based on $\mathcal{P}_s^n$, we define the 2D-SE detachment cost of group $o$ as the increment in 2D-SE when detaching it from its cluster. Suppose $o \in \mathcal{C}_j$, we construct a new partition

$$\mathcal{P}_{s,-o}^n = (\mathcal{P}_s^n \setminus \{\mathcal{C}_j\}) \cup \{\mathcal{C}_j \setminus \{o\}\} \cup \{\{o\}\}, \tag{7}$$

and the corresponding 2D-SE increment is

$$\Delta\mathcal{H}_{s,o}^n = \mathcal{H}^{(2)}(\mathcal{P}_{s,-o}^n) - \mathcal{H}^{(2)}(\mathcal{P}_s^n). \tag{8}$$

For the detailed derivation of Eq. 8, please refer to literature (Xian et al., 2025; Li et al., 2026a). In SEP, a larger $\Delta\mathcal{H}_{s,o}^n$ indicates that group $o$ is less replaceable by other groups in the current coding structure, and detaching it from the current cluster would cause a larger increase in the global coding cost. Since groups within the same cluster are highly redundant, for each sample $n$ we keep only the group with the largest $\Delta\mathcal{H}_{s,o}^n$ in each cluster, which preserves the shared information with minimal loss:

$$r_{s,o}^n = \mathbb{I}[o = \arg\max_{o' \in \mathcal{C}_s^n(o)} \Delta\mathcal{H}_{s,o'}^n], \tag{9}$$

where $\mathcal{C}_s^n(o)$ denotes the cluster containing group $o$ in the partition $\mathcal{P}_s^n$, and $r_{s,o}^n$ indicates whether the group $o$ at stage $s$ for sample $n$ is retained. To make pruning decisions consistent across samples, we compute the selection frequency over the sample pool and retain groups above a threshold:

$$p_s(o) = \frac{1}{N}\sum_{n=1}^{N} r_{s,o}^n, \quad \mathcal{S}_s = \{o \mid p_s(o) \geq \rho\}, \tag{10}$$

where $p_s(o)$ is the selection frequency of group $o$ at stage $s$, $N$ is the pool size, $\rho \in (0, 1]$ is the frequency threshold, and $\mathcal{S}_s$ is the final set of retained groups at stage $s$. Finally, we prune the unselected channel groups stage by stage and perform lightweight fine-tuning to recover performance.

Overall, SEP bounds capacity growth while maintaining stable restoration performance as tasks accumulate. By defining channel group necessity through the change in optimal information coding cost, SEP establishes an explicit and interpretable correspondence between pruning decisions and representation redundancy, yielding a reproducible capacity control mechanism for long-term open-world evolution.

### 3.3. Sub-degradation Knowledge Mining Module

As shown in Figure 2(C), TIR degradations often occur in compound forms, where multiple sub-degradations are coupled nonlinearly. Although historical sub-degradation branches contain transferable priors, compound degradations are not simple linear superpositions; thus, naive concatenation/averaging/fixed-weight fusion may cause negative transfer. Consequently, the key is sample-adaptive retrieval and reorganization of transferable components from historical representations, rather than indiscriminate reuse. We propose the **Sub-degradation Knowledge Mining Module** (SKMM), which treats historical sub-degradation representations as a retrievable knowledge bank and operates on the final-stage output features. Given compound feature $\mathbf{F}^c \in \mathbb{R}^{C \times H \times W}$ and historical sub-degradation features $\{\mathbf{F}_i^h\}_{i=1}^K$ (where $K$ is the number of sub-degradations), we first align them in a shared latent space:

$$\hat{\mathbf{F}}^c = \mathcal{K}_p^c(\mathbf{F}^c), \quad \mathbf{F}^h = \mathcal{K}_p^h\Big(\text{Cat}(\{\mathbf{F}_i^h\}_{i=1}^K)\Big), \tag{11}$$

where $\hat{\mathbf{F}}^c, \mathbf{F}^h \in \mathbb{R}^{M \times H \times W}$, and $\mathcal{K}_p^c$ and $\mathcal{K}_p^h$ are convolutional projections. This alignment enables relevance estimation in a shared latent space, mitigating distribution shifts across branches. Since compound-degradation coupling varies across samples, we use global descriptors of $\hat{\mathbf{F}}^c$ and $\mathbf{F}^h$ to predict a sample-adaptive low-rank mining matrix that captures transferable channel relations. A lightweight hypernetwork $\Phi(\cdot)$ outputs two low-rank factors:

$$[\mathbf{U}, \mathbf{V}] = \Phi\Big(\text{Cat}(\text{GAP}(\hat{\mathbf{F}}^c); \text{GAP}(\mathbf{F}^h))\Big), \tag{12}$$

where $\text{GAP}(\cdot)$ is global average pooling, and $\mathbf{U}, \mathbf{V} \in \mathbb{R}^{M \times R}$ with $R \ll M$. This yields a sample-adaptive low-rank mining matrix $\mathbf{A}$:

$$\mathbf{A} = \sigma(\mathbf{U}\mathbf{V}^\top) \in \mathbb{R}^{M \times M}, \tag{13}$$

where $\sigma(\cdot)$ is the Sigmoid. This low-rank factorization models cross-channel relations with low overhead, without learning a full $M \times M$ matrix. Using $\mathbf{A}$, we retrieve and reorganize historical knowledge via channel mixing and inject the mined components as compensatory residuals into the compound branch:

$$\mathbf{F}^{mix} = [\sum_{j=1}^{M} \mathbf{A}_{i,j}\, \mathbf{F}_j^h]_{i=1}^M, \quad \tilde{\mathbf{F}}^c = \mathbf{F}^c + \mathcal{K}_r(\mathbf{F}^{mix}), \tag{14}$$

where $\mathbf{F}_j^h$ denotes the $j$-th channel feature map of $\mathbf{F}^h$, $\mathbf{F}^{mix} \in \mathbb{R}^{M \times H \times W}$ is the mixed residual feature, $\mathcal{K}_r(\cdot)$ is the residual mapping, and $\tilde{\mathbf{F}}^c$ is the enhanced feature.

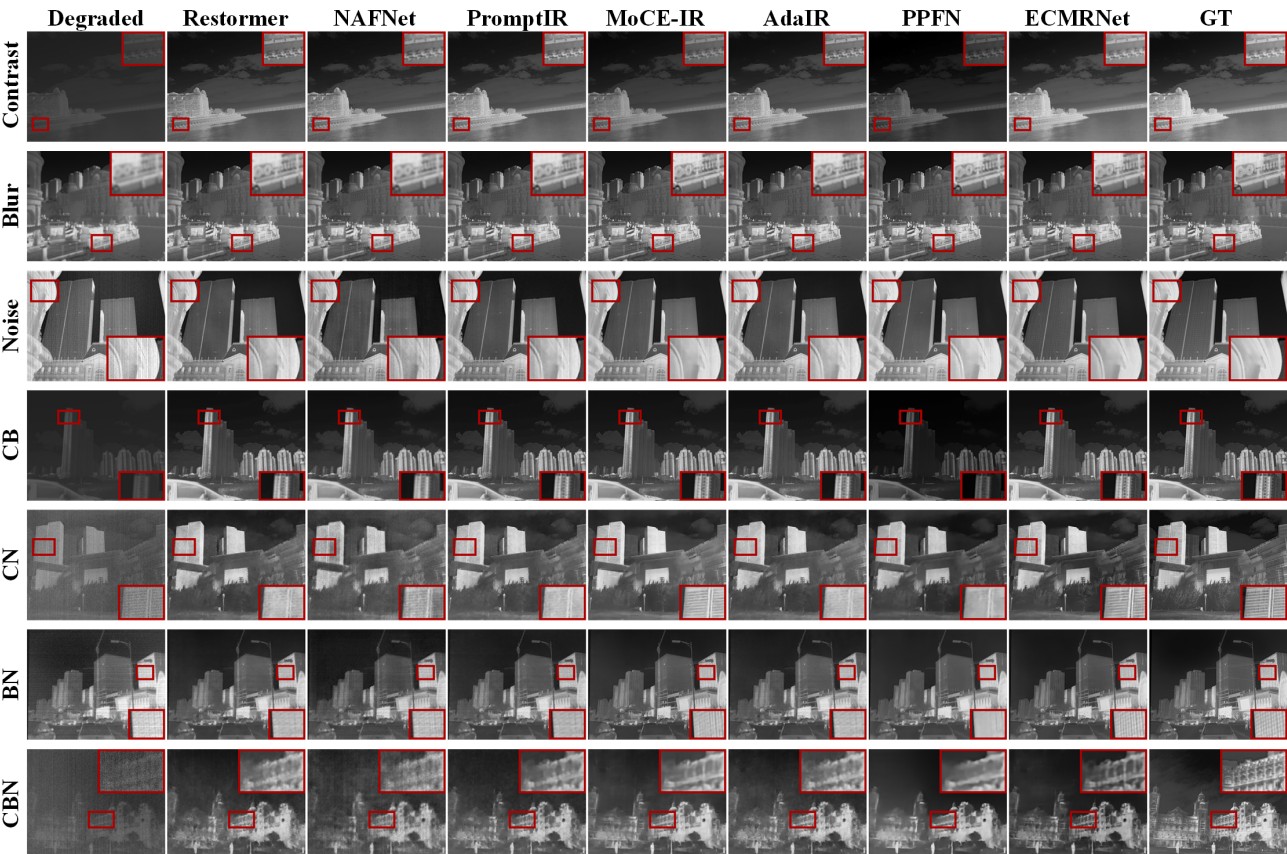

*Figure 3.* Qualitative comparison on the HM-TIR dataset. More qualitative results are provided in the Appendix C.

Importantly, SKMM does not assume all historical knowledge should be reused; instead, it selectively weights channels via sample-adaptive low- rank mining matrix **A** and injects only transferable components, without requiring extra sub-degradation supervision. This enables the compound branch to continually benefit from historical priors at low cost, improving stability and generalization under compound degradations.

## 4. Experiments

### 4.1. Experimental Setups

**Datasets and Evaluation Metrics.** We construct an incremental degradation task sequence on HM-TIR (Liu et al., 2025a), consisting of three single degradations—low contrast (C), blur (B), and noise (N)—and four compound degradations (CB, CN, BN, and CBN). For each task, we follow the degradation synthesis pipeline of PPFN (Liu et al., 2025a) to generate the corresponding degraded samples, and randomly split HM-TIR into training and test sets with an 8:2 ratio to train and evaluate ECMRNet. We further conduct cross-dataset evaluation on M$^3$FD (Liu et al., 2022) using the same degradation protocol to assess generalization. In addition, to examine real-world applicability, we

test on three real-world datasets: EN (Kuang et al., 2019), TIR100 (He et al., 2018), and AWMM (Li et al., 2026b). For evaluation, we report PSNR and SSIM (Wang et al., 2004) on synthetic data, and use MUSIQ (Ke et al., 2021) and CLIP-IQA (Wang et al., 2023) as no-reference perceptual quality metrics on real data.

**Comparison Methods.** We compare ECMRNet with three generic image restoration networks, Restormer (Zamir et al., 2022), NAFNet (Chen et al., 2022), and PromptIR (Potlapalli et al., 2023); two visible all-in-one restoration models, MoCE-IR (Zamfir et al., 2025) and AdaIR (Cui et al., 2025); and a TIR compound-degradation restoration method, PPFN (Liu et al., 2025a). We further include four representative continual-learning paradigms, LwF (Li & Hoiem, 2017), EWC (Kirkpatrick et al., 2017), SI (Zenke et al., 2017), and MAS (Aljundi et al., 2018). For fair comparison, all continual-learning baselines are implemented on top of the same NAFNet backbone.

**Implementation Details.** All experiments are implemented in PyTorch and run on an 4090D GPU with 48 GB of VRAM. Training samples are $256 \times 256$ patches extracted from each image using sliding-window cropping. We optimize an $\ell_1$ loss with Adam using a batch size of 24. The learning rate is initialized to $1 \times 10^{-4}$ and decayed to $1 \times 10^{-6}$ via cosine

*Table 1.* Quantitative comparison on the HM-TIR and M³FD datasets. The best and second-best performances for each metric are highlighted with Red and Blue backgrounds, respectively.

| | Degradation | Contrast (C) | | Blur (B) | | Noise (N) | | CB | | CN | | BN | | CBN | | Avg. | |
|---|---|---|---|---|---|---|---|---|---|---|---|---|---|---|---|---|---|
| | Metric | PSNR | SSIM | PSNR | SSIM | PSNR | SSIM | PSNR | SSIM | PSNR | SSIM | PSNR | SSIM | PSNR | SSIM | PSNR | SSIM |
| **HM-TIR** | Restormer | 33.66 | .974 | 36.72 | .948 | 26.88 | .889 | 31.23 | .916 | 21.60 | .799 | 25.77 | .821 | 21.33 | .753 | 28.17 | .871 |
| | NAFNet | 30.56 | .945 | 34.64 | .926 | 25.43 | .845 | 29.38 | .891 | 20.35 | .721 | 24.40 | .774 | 19.94 | .669 | 26.39 | .824 |
| | PromptIR | 33.35 | .972 | 37.20 | .952 | 26.66 | .897 | 31.66 | .926 | 22.13 | .809 | 26.47 | .831 | 21.82 | .764 | 28.47 | .879 |
| | MoCE-IR | 33.50 | .973 | 39.86 | .969 | 27.31 | .910 | 30.57 | .941 | 22.08 | .827 | 26.44 | .867 | 21.64 | .779 | 28.77 | .895 |
| | AdaIR | 33.71 | .974 | 36.97 | .952 | 26.91 | .893 | 31.80 | .927 | 22.26 | .803 | 25.92 | .826 | 21.29 | .757 | 28.41 | .876 |
| | PPFN | 32.58 | .931 | 39.80 | .969 | 29.41 | .919 | 27.59 | .865 | 22.83 | .793 | 25.58 | .838 | 22.11 | .775 | 28.56 | .870 |
| | LwF | 23.26 | .848 | 23.50 | .815 | 22.76 | .829 | 22.33 | .808 | 19.87 | .757 | 22.61 | .782 | 19.55 | .731 | 22.12 | .796 |
| | EWC | 21.21 | .799 | 22.01 | .751 | 23.11 | .733 | 21.11 | .756 | 20.34 | .686 | 22.68 | .682 | 19.87 | .640 | 21.48 | .721 |
| | MAS | 34.79 | .971 | 21.07 | .749 | 17.95 | .378 | 30.48 | .888 | 17.42 | .285 | 17.70 | .321 | 16.99 | .232 | 22.34 | .546 |
| | SI | 23.34 | .826 | 22.24 | .789 | 22.15 | .789 | 23.59 | .790 | 21.17 | .755 | 22.12 | .737 | 19.61 | .708 | 22.03 | .771 |
| | **ECMRNet** | 37.46 | .989 | 41.72 | .974 | 27.88 | .924 | 34.27 | .956 | 22.49 | .839 | 26.86 | .873 | 21.82 | .794 | 30.36 | .907 |
| **M³FD** | Restormer | 29.01 | .963 | 40.33 | .976 | 26.20 | .910 | 28.19 | .939 | 20.97 | .847 | 25.81 | .885 | 19.80 | .826 | 27.19 | .907 |
| | NAFNet | 29.82 | .961 | 39.16 | .970 | 24.41 | .860 | 29.18 | .937 | 19.62 | .758 | 24.18 | .830 | 19.29 | .731 | 26.52 | .864 |
| | PromptIR | 28.59 | .965 | 40.69 | .977 | 26.44 | .916 | 27.76 | .940 | 21.26 | .857 | 25.98 | .892 | 19.91 | .836 | 27.23 | .912 |
| | MoCE-IR | 29.11 | .970 | 41.71 | .980 | 26.39 | .934 | 27.83 | .952 | 20.82 | .861 | 26.21 | .917 | 20.05 | .842 | 27.44 | .922 |
| | AdaIR | 28.99 | .963 | 40.10 | .975 | 26.29 | .914 | 28.44 | .943 | 20.76 | .849 | 25.86 | .889 | 20.19 | .828 | 27.23 | .909 |
| | PPFN | 29.46 | .919 | 41.36 | .980 | 26.41 | .932 | 26.75 | .892 | 20.91 | .856 | 25.67 | .901 | 20.31 | .843 | 27.27 | .903 |
| | LwF | 21.72 | .889 | 23.19 | .858 | 22.51 | .848 | 22.06 | .872 | 20.84 | .823 | 22.55 | .832 | 19.85 | .801 | 21.82 | .846 |
| | EWC | 18.57 | .799 | 20.58 | .768 | 22.02 | .729 | 18.64 | .782 | 19.63 | .723 | 21.88 | .705 | 19.25 | .697 | 20.08 | .743 |
| | MAS | 31.63 | .968 | 21.07 | .786 | 17.59 | .301 | 31.54 | .942 | 16.98 | .225 | 17.43 | .270 | 16.77 | .201 | 21.86 | .527 |
| | SI | 20.87 | .848 | 20.79 | .815 | 21.12 | .788 | 21.16 | .834 | 20.97 | .796 | 21.24 | .770 | 19.42 | .772 | 20.79 | .803 |
| | **ECMRNet** | 33.28 | .982 | 44.12 | .983 | 26.76 | .939 | 31.14 | .962 | 21.03 | .866 | 26.66 | .921 | 20.39 | .850 | 29.06 | .929 |

annealing. For each task, we train for 100 epochs and then fine-tune for 20 epochs after pruning. In SEP, we use a sample pool of $N = 300$ images randomly drawn from the training set, with the frequency threshold set to $\rho = 0.1$. In SKMM, the dimensions of the low-rank factors are set to $M=32$ and $R=8$. More architectural details of proposed ECMRNet and Baseline implementations are provided in the Appendix A.

### 4.2. Results on Incremental Degradation Tasks

We construct a 7-task incremental degradation sequence on both HM-TIR and M³FD, and report quantitative results in Table 1. Overall, ECMRNet achieves consistent gains across all tasks, improving the average SSIM by 0.01 and PSNR by 1.61 dB, which validates its stable advantage for open-world continual degradation restoration. The qualitative comparisons in Figure 3 further show that ECMRNet produces more natural contrast enhancement, recovers sharper details under blur, and suppresses noise-induced artifacts in single-degradation settings, while preserving clearer structures and edges under compound degradations.

These improvements are largely attributed to the limitations of all-in-one restoration models (Restormer, NAFNet, PromptIR, MoCE-IR, AdaIR, and PPFN), which rely on a single shared parameter set to cover diverse degradations. Such full parameter sharing is prone to cross-degradation gradient interference, thereby capping performance. No-

tably, these methods typically require retraining or global fine-tuning as new degradations emerge, making continual adaptation costly. In contrast, continual-learning baselines (LwF, EWC, MAS, and SI) alleviate forgetting via regularization or distillation; however, in TIR restoration where degradations differ substantially, they often face a dual issue: old tasks can still be disturbed, while new tasks are overly constrained and thus underfit, resulting in unstable overall performance. Benefiting from strict parameter isolation, controllable model growth, and adaptive historical knowledge mining, ECMRNet achieves a better performance–complexity trade-off and continually adapts to emerging degradations in open-world settings.

*Table 2.* Quantitative comparison on the Real-world datasets. 'CLIP' indicates CLIP-IQA. The best and second-best performances for each metric are highlighted with Red and Blue backgrounds, respectively.

| Dataset | EN (CB) | | TIR100 (BN) | | AWMM (CBN) | |
|---|---|---|---|---|---|---|
| Metric | MUSIQ | CLIP | MUSIQ | CLIP | MUSIQ | CLIP |
| Ori. TIR | 65.95 | .278 | 61.10 | .267 | 55.71 | .174 |
| Restormer | 68.63 | .294 | 61.50 | .259 | 56.80 | .183 |
| NAFNet | 65.55 | .269 | 62.32 | .252 | 56.16 | .191 |
| PromptIR | 68.27 | .299 | 61.49 | .266 | 56.34 | .187 |
| MoCE-IR | 66.81 | .303 | 61.66 | .294 | 57.37 | .192 |
| AdaIR | 68.49 | .293 | 61.81 | .265 | 56.30 | .185 |
| PPFN | 67.70 | .305 | 61.69 | .276 | 57.17 | .232 |
| **ECMRNet** | 70.89 | .320 | 62.66 | .338 | 57.50 | .236 |

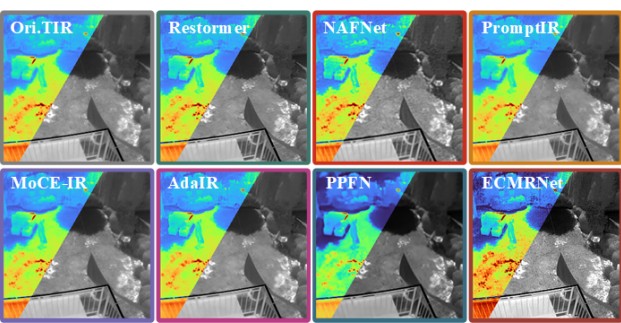

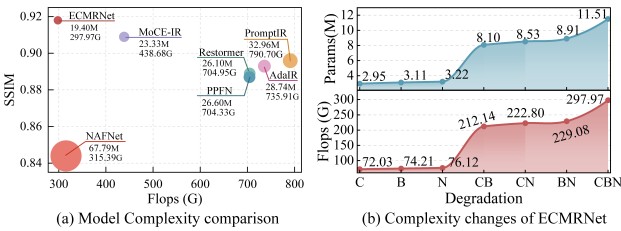

*Figure 4.* Qualitative comparison on the EN dataset. More qualitative results are provided in the Appendix C.

### 4.3. Results on Real-world TIR

To evaluate the real-world generalization of ECMRNet, we conduct experiments on three TIR datasets: EN (low contrast + blur), TIR100 (blur + noise), and AWMM (low contrast + blur + noise). As reported in Table 2, ECMRNet achieves the best results on MUSIQ and CLIP-IQA across all three datasets, indicating higher perceptual quality and stronger semantic consistency. The visual comparisons in Figure 4 further show that ECMRNet enhances global contrast while better recovering structural edges and fine details, and suppresses over-smoothing, leading to more natural, sharp, and consistent results. Overall, ECMRNet maintains a stable advantage in real-world evaluation, demonstrating robust generalization for open-world TIR restoration.

### 4.4. Model Complexity Analysis

Figure 5(a) compares the inference complexity of ECMRNet with competing methods under the same setting ($640 \times 512$). ECMRNet achieves the highest SSIM with only 19.40M parameters, while its maximum inference cost is 297.97G FLOPs, indicating that it attains better restoration quality with substantially reduced computation. Figure 5(b) further reports the activated parameter count and FLOPs of ECMRNet under different degradations. Benefiting from SCGE-based group-wise structural evolution and SEP-based redundancy pruning, the inference cost can be adaptively adjusted according to degradation complexity. For single degradations, ECMRNet activates only 2.95M–3.22M parameters, corresponding to 72.03G–76.12G FLOPs. For compound

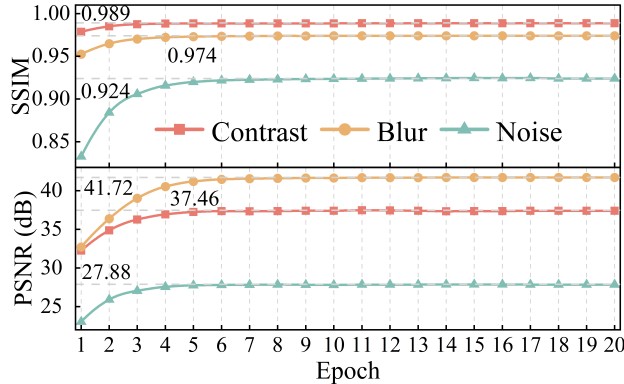

(a) Model Complexity comparison  (b) Complexity changes of ECMRNet

*Figure 5.* Efficiency and scalability of ECMRNet.

*Table 3.* Ablation studies on the SCGE, SEP, and SKMM. * indicates average performance under compound degradations.

| | SCGE | SEP | SKMM | Params (M) | FLOPs (G) | Avg. Metrics | |
|---|---|---|---|---|---|---|---|
| | | | | | | PSNR | SSIM |
| 1 | ✗ | ✗ | ✗ | 4.07 | 97.46 | 26.21 | .851 |
| 2 | ✓ | ✗ | ✓ | 27.51 | 396.97 | 30.51 | .908 |
| 3* | ✓ | ✓ | ✗ | 18.99 | 76.12 | 25.44 | .853 |
| 4* | ✓ | ✓ | ✓ | 19.40 | 297.97 | 26.36 | .866 |
| 5 | ✓ | ✓ | ✓ | 19.40 | 297.97 | 30.36 | .907 |

degradations, the model additionally activates historical sub-degradation branches to support SKMM-based knowledge mining, increasing the activated parameters and computation to 8.10M–11.51M and 212.14G–297.97G FLOPs, respectively, to meet stronger restoration demands. Overall, ECMRNet supports continual degradation adaptation with controllable capacity growth while maintaining efficient and bounded inference complexity.

### 4.5. Ablation Studies

We conduct ablation studies on HM-TIR to evaluate the effectiveness and necessity of SCGE, SEP, and SKMM. Results are reported in Table 3, with additional ablations and analyses provided in the Appendix D.

**Effectiveness of SCGE**. As shown in the first row of Table 3, removing SCGE also disables SEP and SKMM, leading to a substantial performance drop (PSNR/SSIM decrease by 4.15 dB/0.056). This is because SCGE is the key mechanism that introduces task-specific incremental capacity for continual degradation learning. Without SCGE, the model must rely on a single shared backbone to fit different degradations, losing the ability to continually adapt to emerging degradations.

*Figure 6.* Model Convergence in fine-tuning stage

**Effectiveness of SEP**. From Table 3, SEP preserves restoration quality while significantly reducing complexity: PSNR/SSIM changes by only 0.15 dB/0.001, whereas parameters and FLOPs are reduced by 29.5% and 24.9%,

respectively. This validates the effectiveness of SEP based on 2D-SE minimization, which removes redundant channel groups with negligible information loss and keeps model growth controllable. Moreover, Figure 6 shows that the lightweight fine-tuning after pruning recovers over 90% of the performance within the first three epochs and stabilizes after about five epochs. This indicates that SEP enables an efficient "compress–recover" closed-loop optimization during continual learning with little additional cost.

**Effectiveness of SKMM.** As compared in rows 3–4 of Table 3, incorporating SKMM improves the average compound-degradation performance (PSNR/SSIM by 0.92 dB/0.013), demonstrating clear gains from historical knowledge mining. The inference cost increases accordingly because SKMM additionally activates sub-degradation-related branches for retrieval and fusion; however, this overhead is only triggered on demand under compound degradations, matching the higher restoration difficulty.

## 5. Conclusion

We propose ECMRNet, an Expandable, Compressible, and Mineable Restoration Network for open-world TIR restoration. ECMRNet introduces Stage-wise Channel Group Expansion (SCGE) to enable interference-free continual degradation adaptation via strict group-level parameter isolation. To prevent unbounded growth as tasks accumulate, Structural Entropy Pruning (SEP) removes redundant channel groups using 2D structural-entropy minimization, yielding controllable and interpretable capacity evolution. For compound degradations, Sub-degradation Knowledge Mining (SKMM) selectively retrieves and recombines transferable components from historical sub-degradation representations via sample-adaptive low-rank channel mixing to enhance restoration. Experiments on incremental tasks and real-world datasets show that ECMRNet consistently outperforms strong baselines with lower complexity, demonstrating robust scalability in open-world settings.

## Acknowledgements

This work was supported in part by the National Natural Science Foundation of China(62571222, 62276120, 62576163, 62161015), and the Yunnan Fundamental Research Projects (202501AS070123, 202301AV070004).

## Impact Statement

This paper presents work whose goal is to advance the field of Computer Vision. There are many potential societal consequences of our work, none which we feel must be specifically highlighted here.

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

## A. Detailed Architecture and Comparison Method Implementation

**Detailed architecture of ECMRNet.** To balance performance and efficiency, ECMRNet uses a single $3 \times 3$ standard convolution for the feature extractor (FE), bottleneck (BN), and reconstruction module (RM), with the initial channel width set to 32. Within each stage, we use $1 \times 1$ standard convolutions at the entrance and exit for inter-group mixing, while the stage body stacks multiple group residual blocks (GResBlocks) with a $3 \times 3$ group-convolution kernel. The network contains six stages (from shallow to deep and back), with channel widths $\{32, 128, 512, 512, 128, 32\}$ and the corresponding numbers of groups $\{8, 16, 32, 32, 16, 8\}$ (the channel number in each stage is divisible by its group number). In SKMM, both the projection convolutions and the residual mapping use $3 \times 3$ convolutions, and the hypernetwork consists of two linear layers with GELU activations.

**Baseline implementation.** For fair comparison, we retrain all competing methods under the same experimental environment and data splits, including Restormer, NAFNet, PromptIR, MoCE, AdaIR, PPFN, and continual-learning baselines (LwF, EWC, MAS, and SI). PPFN is trained strictly following its original settings, while the other methods are trained with an $\ell_1$ loss and the Adam optimizer, using a batch size ranging from 4 to 24. The initial learning rate is $1 \times 10^{-4}$ and is decayed to $1 \times 10^{-6}$ using cosine annealing, with a total of 100 epochs. During training, we randomly crop $256 \times 256$ patches from input images and apply random flipping for data augmentation. For each iteration, we randomly sample one degradation from the seven-task set and synthesize the corresponding degraded sample online using the associated degradation pipeline.

*Table 4.* Quantitative comparison with DA-CLIP and DiffUIR on the HM-TIR dataset. The best and second-best performances for each metric are highlighted with Red and Blue backgrounds, respectively.

| Degradation | Contrast (C) | | Blur (B) | | Noise (N) | | CB | | CN | | BN | | CBN | | Avg. | |
|---|---|---|---|---|---|---|---|---|---|---|---|---|---|---|---|---|
| Metric | PSNR | SSIM | PSNR | SSIM | PSNR | SSIM | PSNR | SSIM | PSNR | SSIM | PSNR | SSIM | PSNR | SSIM | PSNR | SSIM |
| DA-CLIP | 15.63 | 0.714 | 29.39 | 0.830 | 21.51 | 0.561 | 17.09 | 0.695 | 16.85 | 0.432 | 20.74 | 0.494 | 16.68 | 0.408 | 19.70 | 0.591 |
| DiffUIR | 12.92 | 0.661 | 34.12 | 0.916 | 20.65 | 0.760 | 17.25 | 0.746 | 16.50 | 0.385 | 20.09 | 0.676 | 16.32 | 0.333 | 19.69 | 0.640 |
| **ECMRNet** | 37.46 | .989 | 41.72 | .974 | 27.88 | .924 | 34.27 | .956 | 22.49 | .839 | 26.86 | .873 | 21.82 | .794 | 30.36 | .907 |

## B. More Quantitative Results

To provide a more comprehensive evaluation, we additionally compare ECMRNet with two recent diffusion-related restoration methods, DA-CLIP (Luo et al., 2024) and DiffUIR (Zheng et al., 2024), on the HM-TIR dataset. For a fair comparison, both methods are tested using their official implementations and released model weights. As shown in Table 4, ECMRNet achieves consistently better performance than these baselines across all tasks. While DA-CLIP and DiffUIR have demonstrated strong performance in visible image restoration, their effectiveness is limited when directly transferred to TIR restoration. This is mainly because TIR images exhibit imaging mechanisms and degradation patterns that differ substantially from those of visible images.

## C. More Qualitative Results

We provide additional qualitative comparisons on the M³FD dataset and three real-world datasets (EN, TIR100, and AWMM), as shown in Figure 7 and Figure 8.

**Results on M³FD.** As observed in Figure 7, although M³FD differs from HM-TIR in scene distribution and texture details, ECMRNet still delivers more stable restoration across different degradations. Compared with competing methods, ECMRNet enhances global contrast while better preserving the continuity of structural edges and the consistency of fine-grained textures, producing more natural and cleaner visual results.

**Results on real-world datasets.** Figure 8 presents comparisons on EN (dominated by low contrast and blur), TIR100 (blur and noise), and AWMM (coupled low contrast, blur, and noise). Most competing methods are less stable under real degradations: on the one hand, contrast enhancement can be insufficient or overly stretched, leading to an imbalanced brightness distribution; on the other hand, denoising/deblurring may leave stripe residues, amplify granular artifacts, or over-smooth structural details. In contrast, ECMRNet shows consistent advantages across all three datasets: it improves global contrast while effectively suppressing noise-induced artifacts and better recovering recognizable edges and local details, yielding clearer and more natural visual quality. This observation is consistent with ECMRNet's leading MUSIQ and CLIP-IQA scores in Table 2, further validating its robustness and generalization in real-world open settings.

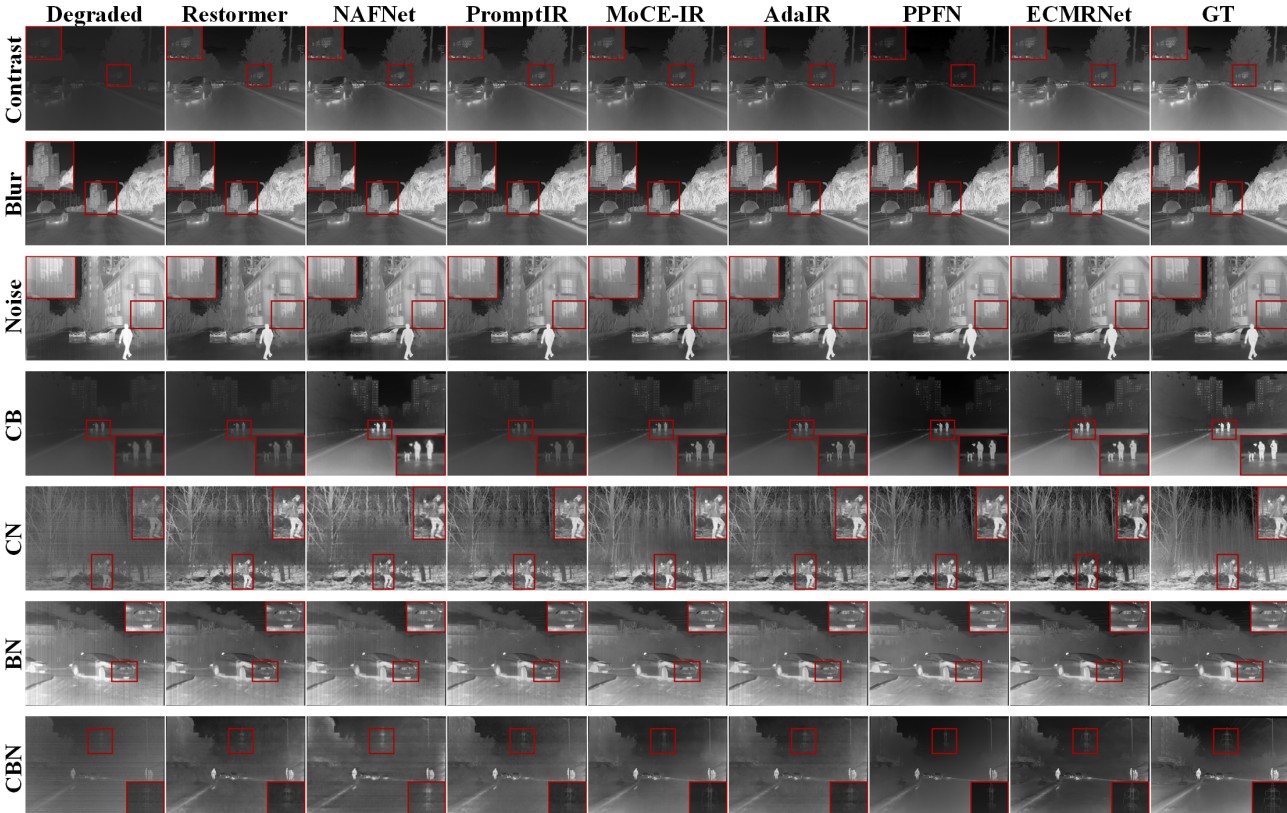

*Figure 7.* Additional qualitative comparison on the M$^3$FD dataset.

## D. Additional Ablation Studies

We further analyze two key hyperparameters in SEP: the sample pool size $N$ and the frequency threshold $\rho$. We report the post-pruning inference complexity (Params/FLOPs) and restoration performance (PSNR/SSIM) on three single-degradation tasks (low contrast, blur, and noise), with results shown in Table 5 and Table 6. In addition, we compare SEP with two classical pruning strategies, Magnitude Pruning (MP) (Han et al., 2015) and Sparsity-Regularized Pruning (SRP). For SRP, we introduce learnable group-wise gates and apply an $\ell_1$ sparsity penalty to them, following the spirit of Network Slimming (Liu et al., 2017). Finally, we evaluate the stability of SEP under a longer and more challenging task sequence.

**The impact of** $N$**.** Table 5 shows that SEP is not sensitive to the sample pool size $N$. When $N$ increases from 100 to 400, PSNR/SSIM on all three degradations remains nearly unchanged, and the pruned Params/FLOPs vary only slightly. This is because the frequency estimate $p(o)$ becomes more stable as $N$ grows, while under a fixed threshold $\rho$, a group must be selected consistently across more samples to be retained, which effectively strengthens the requirement of cross-sample agreement and leads the retained set to converge. Therefore, SEP can provide reliable estimates even with a relatively small sample pool.

*Table 5.* The impact of different Sample pool size $N$ on model performance. The best and second-best performances for each metric are highlighted with Red and Blue backgrounds, respectively.

| Degradation | Contrast | | | | Blur | | | | Noise | | | |
|---|---|---|---|---|---|---|---|---|---|---|---|---|
| Metric | FLOPs | Params | PSNR | SSIM | FLOPs | Params | PSNR | SSIM | FLOPs | Params | PSNR | SSIM |
| $N = 100$ | 71.48G | 2.95M | 37.46 | 0.989 | 72.56G | 3.05M | 41.72 | 0.974 | 76.66G | 3.27M | 27.88 | 0.924 |
| $N = 200$ | 73.12G | 2.97M | 37.46 | 0.989 | 73.66G | 3.10M | 41.72 | 0.974 | 78.85G | 3.38M | 27.88 | 0.924 |
| $N = 300$ | 72.03G | 2.95M | 37.46 | 0.989 | 74.21G | 3.11M | 41.72 | 0.974 | 76.12G | 3.22M | 27.88 | 0.924 |
| $N = 400$ | 73.67G | 3.06M | 37.47 | 0.989 | 72.29G | 3.83M | 41.76 | 0.974 | 75.57G | 3.17M | 27.88 | 0.924 |

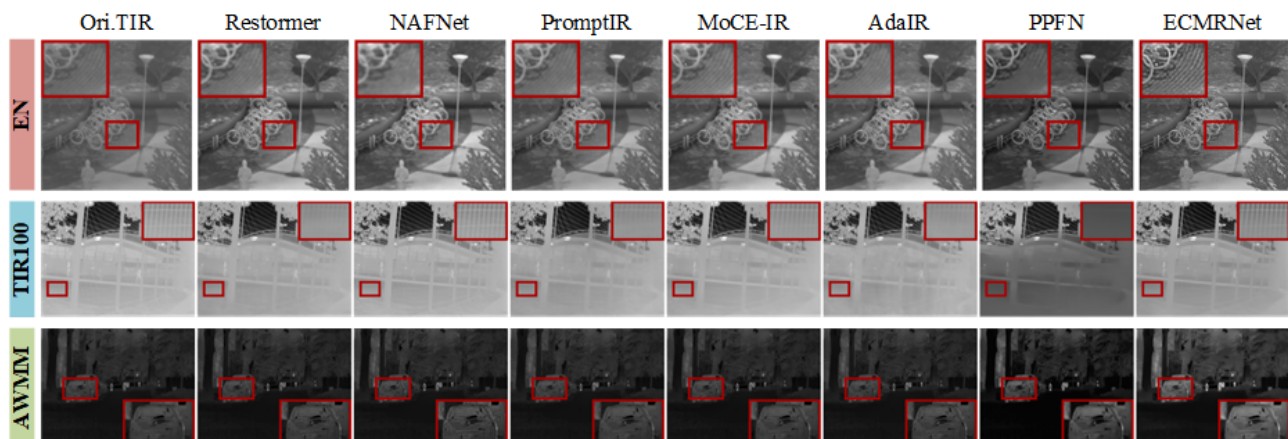

*Figure 8.* Additional qualitative comparison on real-world datasets (EN, TIR100, and AWMM).

**The impact of $\rho$.** Table 6 illustrates the direct effect of $\rho$ on the performance–complexity trade-off. With a small $\rho$ (e.g., 0.05), more low-contribution or sample-specific groups are retained; this may slightly improve some metrics but yields limited overall benefit, while increasing Params/FLOPs and reducing pruning efficiency. Conversely, with an overly large $\rho$ (e.g., 0.15/0.20), only a few high-frequency groups are kept, further reducing complexity but potentially pruning groups that remain important for certain degradations, leading to more noticeable PSNR/SSIM drops. Empirically, $\rho=0.10$ achieves a better trade-off across the three degradations by reducing complexity with negligible performance loss. Overall, increasing $\rho$ makes Params/FLOPs decrease approximately monotonically, but overly large thresholds cause clear performance degradation.

*Table 6.* The impact of different frequency thresholds $\rho$ on model performance. The best and second-best performances for each metric are highlighted with Red and Blue backgrounds, respectively.

| Degradation | | Contrast | | | | Blur | | | | Noise | | |
|---|---|---|---|---|---|---|---|---|---|---|---|---|
| Metric | FLOPs | Params | PSNR | SSIM | FLOPs | Params | PSNR | SSIM | FLOPs | Params | PSNR | SSIM |
| $\rho = 0.05$ | 85.97G | 3.59M | 37.54 | 0.989 | 82.42G | 3.38M | 41.72 | 0.974 | 89.52G | 3.81M | 28.15 | 0.925 |
| $\rho = 0.10$ | 72.03G | 2.95M | 37.46 | 0.989 | 74.21G | 3.11M | 41.72 | 0.974 | 76.12G | 3.22M | 27.88 | 0.924 |
| $\rho = 0.15$ | 60.01G | 2.42M | 37.20 | 0.984 | 61.90G | 2.59M | 41.58 | 0.971 | 61.35G | 2.66M | 27.13 | 0.919 |
| $\rho = 0.20$ | 47.42G | 1.93M | 36.34 | 0.974 | 53.15G | 2.27M | 40.73 | 0.965 | 49.87G | 2.10M | 26.67 | 0.907 |

**Comparison with MP and SRP.** For a fair comparison, MP and SRP adopt a Top-$K$ strategy to retain the same number of channel groups as SEP. We evaluate all strategies on three single-degradation tasks, including low contrast, blur, and noise, without additional fine-tuning. As shown in Table 7, SEP achieves more effective and stable pruning performance overall. In particular, on the blur task, SRP suffers from a significant performance drop, suggesting that it incorrectly removes some critical channel groups. In contrast, SEP evaluates the structural contribution of each channel group through the 2D-SE criterion and further incorporates cross-sample frequency-based selection, leading to more consistent retention decisions. Importantly, SEP does not rely on a fixed pruning ratio; instead, it adaptively removes redundant channel groups according to their structural information contribution.

*Table 7.* Comparison of different pruning strategies without fine-tuning. The best and second-best performances for each metric are highlighted with Red and Blue backgrounds, respectively.

| Strategy | Training | Contrast | | Blur | | Noise | | Avg. | |
|---|---|---|---|---|---|---|---|---|---|
| | | PSNR | SSIM | PSNR | SSIM | PSNR | SSIM | PSNR | SSIM |
| MP | ✗ | 19.53 | 0.879 | 19.44 | 0.882 | 17.72 | 0.749 | 18.90 | 0.837 |
| SRP | ✓ | 23.05 | 0.784 | 5.36 | 0.053 | 18.85 | 0.709 | 15.75 | 0.515 |
| SEP | ✗ | 28.99 | 0.966 | 28.76 | 0.924 | 19.37 | 0.781 | 25.71 | 0.890 |

**The stability of SEP.** To further evaluate the robustness of SEP, we extend the original 7-task sequence by adding two additional tasks on HM-TIR dataset: one single-degradation task (dead pixels), and one more challenging compound-

degradation task consisting of low contrast, blur, noise, and dead pixels (CBND). We then compare the overall performance before and after SEP pruning and fine-tuning on these newly added tasks. As shown in Table 8, the performance gap before and after SEP remains small on both tasks, while the model Params and FLOPs are effectively reduced. These results further demonstrate the robustness and stability of SEP under longer and more complex task sequences.

*Table 8.* Comparison of the model before and after pruning and fine-tuning on newly introduced degradation tasks. The best performance for each metric is highlighted with Red background.

| Using SEP | Dead Pixels | | | | CBND | | | |
|---|---|---|---|---|---|---|---|---|
| | FLOPs | Params | PSNR | SSIM | FLOPs | Params | PSNR | SSIM |
| ✗ | 97.46G | 4.07M | 45.34 | 0.991 | 383.27G | 14.78M | 21.82 | 0.785 |
| ✓ | 56.16G | 2.37M | 45.10 | 0.989 | 355.91G | 13.68M | 21.55 | 0.782 |

## E. Limitations and Future Work

This work studies open-world TIR image restoration from the perspective of continual degradation learning. Specifically, we follow a task-incremental protocol, where new degradation tasks arrive sequentially and the model is required to absorb new degradation knowledge without retraining on previous tasks, while preserving the restoration capability for historical degradations and controlling long-term model growth. Under this setting, the degradation type is assumed to be available at inference for learned tasks. In practical applications, this can be supported by a lightweight degradation recognizer for identifying learned degradation types. Therefore, the main focus of ECMRNet is continual adaptation to evolving degradations, preservation of historical restoration ability, and controlled model evolution, rather than directly addressing arbitrary unseen degradation recognition and restoration under a stronger open-set setting without task boundaries or degradation labels. In future work, it would be valuable to model degradation streams with longer and more complex composition patterns, and to incorporate more real-world data for improving the generalization from synthetic task streams to practical open environments. Meanwhile, extending the model toward stronger open-world scenarios, such as recognizing and restoring previously unseen degradations, remains an important direction for future research.

