# OpenReview forum: "Expandable, Compressible, Mineable: Open-World Thermal Infrared Image Restoration"
_ICML.cc/2026/Conference — ICML 2026 regular_

### Official Review · Reviewer_e3Bg · 2026-03-04

**Soundness:** 3
**Presentation:** 3
**Significance:** 2
**Originality:** 2
**Overall Recommendation:** 4
**Confidence:** 3

**Summary:**

The paper proposes ECMRNet, a framework that unifies continuous degradation learning through a closed-loop process of expansion–compression–mining, aiming to enable adaptation to newly emerging degradations while maintaining controllable model evolution. The proposed pipeline introduces stage-wise channel group expansion (SCGE) for parameter isolation, structural entropy pruning (SEP) to regulate model growth, and a sub-degradation knowledge mining module (SKMM) to improve restoration under compound degradation scenarios. Experiments conducted on both synthetic and real-world TIR datasets show improvements over several existing restoration and continual-learning baselines.

**Compliance With Llm Reviewing Policy:**

Affirmed.

**Key Questions For Authors:**

1. The framework assumes that degradations arrive as clearly defined tasks. However, in real-world scenarios, degradations are often continuous and mixed. It would be helpful if the authors could clarify how the proposed method handles settings where task boundaries are unclear.
2. The group-isolated design effectively prevents interference between tasks, but it may also limit feature sharing across different degradations. Since many restoration tasks share common low-level representations, could the authors discuss the potential trade-off between task isolation and representation sharing?

**Limitations:**

The work does not appear to raise significant societal risks.

**Strengths And Weaknesses:**

Strengths:

1. The proposed ECMRNet architecture presents a systematic design that integrates continual learning principles with image restoration tasks.

2. The expand–compress–mine framework is conceptually clear and provides a coherent strategy for balancing model capacity growth and knowledge reuse.

3. Experimental results on multiple datasets demonstrate consistent performance improvements along with reasonable computational efficiency.

Weaknesses:

1. The method consists of multiple modules, which makes the overall framework relatively complex and may increase the difficulty of reproduction, particularly during the training phase.

2. Given the recent progress in diffusion-based restoration methods, it would be beneficial to include comparisons with representative diffusion-based approaches.

3. The motivation for structural entropy pruning (SEP) is not sufficiently elaborated. Additional comparisons with standard pruning techniques could further strengthen the justification of this design.

---

> ### Author Rebuttal · Authors · 2026-03-30
>
> We sincerely thank you for your thorough review and positive feedback.
> Below we respond to each concern.
>
> ---
> **W1:**
> We agree that ECMRNet contains multiple components and that its training pipeline is relatively complex. However, each component addresses a distinct challenge in open-world TIR restoration: SCGE enables interference-free continual adaptation to newly arriving degradations, SEP controls model growth, and SKMM mines historical sub-degradation knowledge to enhance compound-degradation restoration. Importantly, the training procedure is stage-wise (expand → compress → mine) rather than fully entangled, which improves reproducibility. To further facilitate reproduction, we will release the complete codebase, provide a clearer pipeline description, and add the code link in the revision.
>
> ---
> **W2:**
> Following your suggestion, we additionally evaluate two recent diffusion-related methods, DA-CLIP and DiffUIR, using their official implementations. The results (see table below) show that ECMRNet consistently outperforms both methods across all tasks. Although they are strong recent baselines for visible-image restoration, their effectiveness remains limited when directly transferred to the TIR setting. We will include these results in the revision.
>
> **Table 1.** Quantitative comparison with DA-CLIP and DiffUIR on the HM-TIR dataset.
>
> | Method | C-PSNR | C-SSIM | B-PSNR | B-SSIM | N-PSNR | N-SSIM | CB-PSNR | CB-SSIM | CN-PSNR | CN-SSIM | BN-PSNR | BN-SSIM | CBN-PSNR | CBN-SSIM | Avg.-PSNR | Avg.-SSIM |
> |:------|:------:|:------:|:------:|:------:|:------:|:------:|:-------:|:-------:|:-------:|:-------:|:-------:|:-------:|:--------:|:--------:|:---------:|:---------:|
> | DA-CLIP | 15.63 | 0.714 | 29.39 | 0.830 | 21.51 | 0.561 | 17.09 | 0.695 | 16.85 | 0.432 | 20.74 | 0.494 | 16.68 | 0.408 | 19.70 | 0.591 |
> | DiffUIR | 12.92 | 0.661 | 34.12 | 0.916 | 20.65 | 0.760 | 17.25 | 0.746 | 16.50 | 0.385 | 20.09 | 0.676 | 16.32 | 0.333 | 19.69 | 0.640 |
> | ECMRNet | **37.46** | **0.989** | **41.72** | **0.974** | **27.88** | **0.924** | **34.27** | **0.956** | **22.49** | **0.839** | **26.86** | **0.873** | **21.82** | **0.794** | **30.36** | **0.907** |
>
> C/B/N denote Contrast/Blur/Noise, respectively.
>
> ---
> **W3:**
> SEP is motivated by the long-term model growth and group-level functional redundancy introduced by SCGE expansion. It is therefore not a generic compression add-on, but an adaptive pruning mechanism aligned with the group-wise granularity of ECMRNet. Unlike standard pruning methods using a fixed pruning ratio, SEP evaluates each newly added channel group by measuring the change in 2D SE after removing it from the current optimal coding structure, thereby enabling information-contribution-driven adaptive pruning. To further justify this design, we additionally compare SEP with magnitude pruning and sparsity-regularized pruning. As discussed in our response to **Reviewer EchJ (W2)** , SEP is overall more effective and stable.
>
> ---
> **KQ1:**
> ECMRNet follows the task-incremental protocol, where the degradation type is known at test time. The core question studied here is whether the model can continually adapt to newly emerging degradations without retraining all previous tasks while keeping model growth under control. Experimental results on three real-world compound degradation datasets validate its robustness and effectiveness. We agree that scenarios with unclear task boundaries or continuously evolving degradations are important. Handling such settings would likely require an additional online and expandable degradation identifier/router on top of ECMRNet, which is beyond the main focus of this paper. We will clarify this limitation in the revision.
>
> ---
> **KQ2:**
> We agree that different degradations share common low-level priors, such as edges, local structures, and basic textures. Accordingly, ECMRNet adopts a strategy of shared base representations with isolated incremental adaptation. Specifically, we first learn a degradation-agnostic shared foundation through clean-to-clean self-reconstruction pretraining, and then freeze the feature extractor, bottleneck, and reconstruction module as stable shared representations across tasks. The group-isolated design is then applied only to the incremental adaptation part for newly arriving degradations.The motivation is that, although low-level priors can be shared, the restoration mappings and parameter update directions of different degradations are often inconsistent; under an open-world continual setting, repeatedly updating shared trainable parameters can easily cause cross-task interference and forgetting. Therefore, ECMRNet does not use isolation to reject sharing. Instead, the shared frozen base supports reusable generic representations, while the isolated incremental groups enable stable degradation-specific adaptation. In this sense, ECMRNet adopts a controlled sharing strategy that is better suited to continual restoration.

---

> > ### Author Rebuttal · Reviewer_e3Bg · 2026-04-02
> >
> > Thanks to the authors for their responses. The rebuttal has addressed my questions. I will maintain my rating.

---

> > > ### Author Response · Authors · 2026-04-03
> > >
> > > We sincerely thank you for your careful review and positive feedback. We are glad that our rebuttal has adequately addressed your concerns. We will incorporate the relevant clarifications and additional experimental results into the revised manuscript.
> > >
> > > Best regards,
> > >
> > > The authors of Submission 7877

---

### Official Review · Reviewer_QuvW · 2026-03-09

**Soundness:** 3
**Presentation:** 3
**Significance:** 2
**Originality:** 3
**Overall Recommendation:** 4
**Confidence:** 4

**Summary:**

The paper proposes ECMRNet, a continual-learning framework for open-world thermal infrared (TIR) image restoration that addresses continually evolving single and compound degradations. The method introduces a closed-loop “expand–compress–mine” pipeline:
- Stage-wise Channel Group Expansion (SCGE) adds task-specific capacity via isomorphic group convolutional subspaces with strict parameter isolation.
- Structural Entropy Pruning (SEP) adaptively prunes redundant channel groups by minimizing 2D structural entropy.
- Sub-degradation Knowledge Mining Module (SKMM) dynamically recombines transferable components from historical sub-degradation features via low-rank channel mixing.

Experiments on incremental degradation sequences (synthetic) and several real-world TIR datasets report improvements in restoration quality and a good performance-complexity trade-off.

**Compliance With Llm Reviewing Policy:**

Affirmed.

**Final Justification:**

I appreciate the author's effort to further address my follow-up concerns, especially the additional analysis on the longer task stream.

Overall, I think the paper is quite good, and most of my main concerns have now been addressed to a reasonable extent. The remaining issue is mainly about scope calibration. In particular, I see it more as a question of how broadly one chooses to interpret the current experimental support, rather than as evidence against the core technical contribution itself. **Personally, I do not see the remaining concerns as strong enough to justify rejection.**

I will therefore maintain my current score. If the paper is not accepted this round, I would be happy to see a tighter revision in the future and would likely view it more favorably then.

**Key Questions For Authors:**

1. Your task stream is built from a specific synthetic degradation pipeline. How sensitive are the conclusions to the chosen synthesis operators and parameter ranges? If the degradation family or ordering changes, do you observe similar gains and stability?
2. The paper reports that different degradations activate different subsets/branches. In an open-world setting where degradation labels are not available, what is the concrete test-time routing mechanism (e.g., degradation classifier, confidence-based gating, or always-on execution)? How does routing error affect both performance and the reported adaptive FLOPs?
3. SEP uses a sample pool and a frequency threshold to decide which groups to prune. How robust is SEP on longer task streams, and does pruning ever remove groups that later become important for compound degradations?
4. CL baselines obtain extremely low PSNR on multiple tasks in the main results table. Could you provide training diagnostics and hyperparameter sweeps to rule out under-tuning?

**Limitations:**

The main limitations are unclear robustness beyond the fixed synthetic degradation stream, under-specified test-time routing for unknown degradations, and insufficient baseline validation (pruning and CL tuning).

**Strengths And Weaknesses:**

Strengths:
- The method is clearly modularized with explicit formulations, and ablations support the necessity of each component (e.g., SCGE for continual adaptation, SEP for controlling growth, and SKMM for compound degradations).
- The evaluation covers both incremental synthetic degradation tasks and real-world datasets with no-reference perceptual metrics that helps connect the continual-learning setup to practical scenarios.
- The paper provides an explicit efficiency/scalability analysis: it reports activated parameter/FLOP budgets that vary with degradation complexity (single vs. compound), which helps connect the proposed expand-compress-mine design to deployment-oriented resource constraints.

Weaknesses:
- The “open-world” stream is instantiated as a fixed sequence of synthetic degradations (and their compounds). It is unclear how well this setup matches truly unseen or evolving real degradations beyond the specific synthesis pipeline, and whether conclusions generalize to different degradation families or non-stationary mixtures.
- Test-time routing/activation is not fully specified: while the paper reports that different degradations activate different subsets/branches (and that SKMM is additionally activated for compound degradations), it's not clear how the system decides which path to execute when degradation type is unknown in an open-world setting.
- The pruning component is only ablated internally (with/without SEP) but not compared against standard structured pruning baselines (e.g., magnitude-based channel pruning, sparsity-regularized pruning, etc) under matched pruning ratios, so it is difficult to attribute the compression-quality trade-off specifically to the structural-entropy criterion.
- Although several restoration and continual-learning baselines are included, it is not fully clear if competing continual strategies are finetuned to their best settings for this restoration context, and additional comparisons to other parameter-isolation / modular-adaptation designs in restoration would strengthen the empirical case.

---

> ### Author Rebuttal · Authors · 2026-03-30
>
> We sincerely thank you for your thorough review and constructive feedback.
> Below we address each of the raised concerns in detail.
>
> ---
> **W1:**
> We acknowledge that the degradation stream is built from a fixed synthetic pipeline and thus provides a reproducible approximation of open-world TIR restoration, rather than a full characterization of real-world evolving degradations. We adopt this setting to test, under comparable conditions, whether a model can continually adapt to newly arriving degradations without retraining all previous tasks while keeping model growth controlled. Beyond the synthetic stream, we further evaluate ECMRNet on three real-world datasets, where it also shows clear robustness. We nevertheless agree that broader degradation families, non-stationary mixtures, and unseen real degradation streams remain important future directions.
>
> ---
> **W2&KQ2:**
> ECMRNet follows the task-incremental protocol, where the degradation type is known at test time. Test-time routing is deterministic and based on the task label, rather than predicted by an additional identifier. The core question studied here is whether the model can continually adapt to newly emerging degradations without retraining all previous tasks while keeping model growth under control. If the degradation is unknown, an additional expandable degradation identifier/router would be required, which is beyond the setting of this work. We will clarify this setting in the revision. Under incorrect routing, the model would activate a mismatched restoration path, degrading restoration quality, and the activated FLOPs would also change.
>
> ---
> **W3:**
> Following your suggestion, we compare SEP with magnitude pruning and sparsity-regularized pruning. For fairness, them retain the same number of channel groups as SEP via Top-K . The results (see our response to **Reviewer EchJ (W2)**) show that SEP is overall more effective and stable. this is attributed to SEP’s evaluation of each channel group by its 2D SE contribution to the current optimal coding structure, together with its cross-sample frequency-based selection, which yields more consistent retention decisions.
>
> ---
> **W4&KQ4:**
> Since CL baselines lack official implementations for image restoration, we re-implement them under a unified NAFNet backbone with the same training setting, so that the comparison mainly reflects differences in the CL mechanisms. We further provide training results after each task and key hyperparameter sweeps for them  ([see the anonymous link](https://anonymous.4open.science/r/ECMRNet-4418/R.png)). Results show that these methods face the same trade-off in restoration: stronger constraints hinder new-task adaptation, while weaker constraints lead to more severe forgetting.  Although LwF improves after tuning its loss-balance weight, it still remains clearly below ECMRNet, suggesting that the gap is not mainly caused by obvious under-tuning. We agree that more comparisons with parameter-isolation or modular-adaptation designs would be valuable; however, directly comparable plug-in methods for TIR restoration remain limited, and many alternatives require substantial backbone redesign, making fully fair comparisons difficult.
>
> ---
> **KQ1:**
> The task stream follows the synthetic degradation pipeline of PPFN, with parameter ranges covering mild to severe degradation levels, thus providing a controlled testbed for continual restoration. Under this setting, ECMRNet shows clear advantages on both synthetic streams and three real-world datasets. As discussed in our response to **Reviewer EchJ (W3)**, ECMRNet is insensitive to the degradation order, because each new degradation is learned in newly expanded groups while historical ones remain frozen. We agree that validating more degradation families and synthesis operators would further strengthen the conclusions, and we will discuss this more explicitly in the revision.
>
> ---
> **KQ3:**
> SEP is designed to support long-term capacity control in extended task streams. After each task, it prunes only the groups newly introduced for that task and uses cross-sample frequency statistics to obtain consistent retention decisions. Its role is therefore to preserve critical channel groups while suppressing redundant growth as tasks accumulate. Tab. 3 and Fig. 6 show that SEP yields substantial reductions in parameters and FLOPs with only marginal performance change, supporting its stability in continual degradation restoration. For compound degradations, the key point is that they depend on retained critical historical representations rather than all previously added groups. SEP never modifies the already retained historical groups of earlier tasks, and SKMM selectively retrieves transferable components from historical sub-degradation representations. Therefore, pruning redundant new groups does not fundamentally undermine the knowledge required for later compound degradation restoration.
>
> ---
> **Limitations:**
> Please see the above response.

---

> > ### Author Rebuttal · Reviewer_QuvW · 2026-04-01
> >
> > Thank you for the clarification. I think several concerns are better clarified now, but I still have several questions that are partially resolved.
> >
> > - I appreciate the clarification that the degradation stream is intended as a reproducible approximation. However, this also suggests that the paper’s claims should be phrased more narrowly and aligned more carefully with the actual evaluation setup, since the current “open-world” framing still feels somewhat broader than what is directly validated.
> >
> > - The routing/activation issue is also clarified, but not fully resolved in the stronger open-world sense. The rebuttal makes clear that the current setting is task-incremental, with known degradation labels at test time and deterministic routing. That answers how routing is done in this paper, but it also means that the key difficulty of routing under truly unknown degradations remains outside the current scope.
> >
> > - Regarding SEP and future compound degradations, pruning is applied only to newly introduced groups for the current task, while previously retained historical groups remain unchanged, so the argument is that later compound degradations still rely on preserved critical historical representations. This addresses part of the concern at the design level. However, what still seems insufficient is the long-horizon evidence. The current results mainly show that pruning works well within the present 7-task setup, with small immediate performance loss and quick recovery. That is useful, but it does not yet demonstrate stability over longer and more complex task streams, where repeated expand-prune cycles may gradually remove groups that later become important. More direct quantitative evidence along this axis would make the claim much more convincing.

---

> > > ### Author Response · Authors · 2026-04-03
> > >
> > > Thank you for your acknowledgment and constructive feedback. We appreciate the opportunity to further clarify these points.
> > >
> > > ---
> > > **W1:**
> > > We use a synthetic degradation stream because, under supervised continual restoration, real degraded TIR images from open-world scenarios are usually difficult to obtain together with strictly paired references. Synthetic degradations instead provide controllable and consistent supervision for both training and systematic evaluation. In this paper, open-world refers to the ability to continually adapt to newly emerging degradation types that are not fully known in advance. The synthetic stream thus serves as a reproducible incremental task sequence. Meanwhile, experiments on multiple real-world degraded datasets further support the robustness and generalization of ECMRNet in practical scenarios. In the revision, we will carefully reconsider the use of the term open-world and further clarify the correspondence between the current setting and the actual evaluation scope, so that the presentation is more precisely aligned with the evaluation protocol.
> > >
> > > ---
> > > **W2:**
> > > We agree that automatic routing/activation for truly unknown degradations is not fully resolved by the current paper in the stronger open-world sense. However, the core problem addressed here remains important: how to continually adapt to new degradations without forgetting previous knowledge, while controlling long-term model growth. To our knowledge, this has not been systematically addressed in TIR restoration. For already learned degradations, inference-time routing can be handled by a very lightweight auxiliary classifier. As a preliminary validation, we trained a classifier with two ResBlocks and one classification head, achieving high accuracy on the learned degradation sequence (F1 = 0.995, ACC = 0.981). This suggests that routing is not the main bottleneck within the known degradation set. Therefore, once the degradation type is determined, ECMRNet can serve as a key continual-learning component in an open-world TIR restoration system. We will make this scope and limitation clearer in the revision.
> > >
> > > ---
> > > **W3:**
> > > To further validate SEP under a longer and more complex task stream, we extend the original 7-task sequence with two additional tasks: one single degradation task (dead pixels) and one harder compound task composed of low contrast, blur, noise, and dead pixels. As shown in Table 1, the performance gap before and after SEP pruning and fine-tuning remains very small on both new tasks, further supporting the robustness and stability of SEP.
> > >
> > > **Table 1.** Comparison of the model before and after pruning and fine-tuning on newly introduced degradation tasks.
> > >
> > > | SEP | D FLOPs (G) | D Params (M) | D PSNR | D SSIM | CBND FLOPs (G) | CBND Params (M) | CBND PSNR | CBND SSIM |
> > > |-----|:------------:|:-------------:|:-------:|:-------:|:---------------:|:----------------:|:----------:|:----------:|
> > > | ×   | 97.46 | 4.07 | 45.34 | 0.991 | 383.27 | 14.78 | 21.82 | 0.785 |
> > > | √   | 56.16 | 2.37 | 45.10 | 0.989 | 355.91 | 13.68 | 21.55 | 0.782 |
> > >
> > > C/B/N/D denote Contrast/Blur/Noise/ Dead pixels, respectively.
> > >
> > > In addition, regarding the concern that SEP may remove channel groups that are important for later compound degradations, we would like to further clarify that SKMM that SKMM takes as input the final-stage output features of historical sub-degradation branches, rather than all channel-group features from the intermediate U-Net stages. Therefore, the most direct test is whether these historical representations remain stable after SEP pruning and fine-tuning. For each historical sub-degradation branch, we use future compound-degradation images containing the corresponding sub-degradation as input and compare the final-stage features before and after pruning. As shown in Table 2, the representations remain highly consistent, indicating that SEP largely preserves the historical sub-degradation information required for later compound restoration.
> > >
> > > **Table 2.** Final-stage feature consistency before and after SEP pruning and subsequent fine-tuning. Cosine similarity (Cos) and linear Centered Kernel Alignment (CKA) are used for comparison.
> > > |      | CB Cos | CB CKA | CN Cos | CN CKA | BN Cos | BN CKA | CBN Cos | CBN CKA | CBND Cos | CBND CKA |
> > > |------|:-------:|:-------:|:-------:|:-------:|:-------:|:-------:|:--------:|:--------:|:---------:|:---------:|
> > > | C    | 0.996 | 0.973 | 0.993 | 0.971 | - | - | 0.993 | 0.966 | 0.974 | 0.993 |
> > > | B    | 0.998 | 0.999 | - | - | 0.991 | 0.999 | 0.992 | 0.999 | 0.999 | 0.999 |
> > > | N    | - | - | 0.988 | 0.978 | 0.976 | 0.968 | 0.979 | 0.956 | 0.968 | 0.963 |
> > > | D    | - | - | - | - | - | - | - | - | 0.988 | 0.999 |
> > >
> > > C/B/N/D denote Contrast/Blur/Noise/ Dead pixels, respectively.
> > >
> > > ---
> > >
> > > Thank you again for your time and careful reading. If any part remains unclear, we would be very happy to further clarify it.

---

### Official Review · Reviewer_BLny · 2026-03-12

**Soundness:** 2
**Presentation:** 3
**Significance:** 2
**Originality:** 2
**Overall Recommendation:** 2
**Confidence:** 5

**Summary:**

This paper proposes an open-world TIR restoration method, which designs a parameter isolation manner to prevent catastrophic forgetting for all-in-one image restoration methods. To reduce the accumulation of model parameters as the number of tasks increases, a structural entropy pruning method is also proposed. Experimental results show the effectiveness for the TIR restoration task.

**Compliance With Llm Reviewing Policy:**

Affirmed.

**Final Justification:**

My main concern is that the problem formulation of open-world image restoration for infrared images is not sufficiently well-defined or well-justified, which is also the main contribution of the paper. When the task setting itself is unclear, it becomes difficult to properly assess the validity, significance, and generalizability of the proposed method. More importantly, an imprecise or potentially inappropriate formulation at the problem-definition stage may introduce confusion to the community and risk misleading future research in this direction. For these reasons, I do not believe the paper is ready for acceptance in its current form. Therefore, I maintain my orginal score.

**Key Questions For Authors:**

Please see the Strengths And Weaknesses.

**Limitations:**

The motivation is unclear, and key experiments are missing. Therefore, it is difficult to determine whether this research is effective and meaningful.

**Strengths And Weaknesses:**

Strengths:

1. The paper is easy to follow.

Weaknesses:

1.	The paper fails to clearly clarify the significance of the evolving recovery of TIR images. Specifically, it does not quantify how much old degradation knowledge is forgotten when a new type of degradation emerges, particularly in the context of infrared image restoration.

2.	As the claimed “first study of open-world TIR image restoration”, the paper should provide a clear and formal definition.

3.	What are the key distinctions between all-in-one visible image restoration and all-in-one TIR image restoration under the open-world setting? In my view, this is the motivation of this paper.

4.	Without the formal definition and the clear motivation, the proposed method is merely an incremental improvement. The design for the continual learning is a naïve parameter-isolation method, and the pruning is directly follows existing methods, without new insights for the specific task.

5.	Figure 1 is hard to follow. A straightforward explanation is required, rather than the vague caption.

6.	In continual learning evaluations, upper bound and forgetting rates are critical indicators, but these are not provided in this paper. Therefore, it’s difficult to determine whether the proposed method is effective.

---

> ### Author Rebuttal · Authors · 2026-03-30
>
> We sincerely thank you for your thorough review and feedback.
> Below we address each of the raised concerns in detail.
>
> ---
> **W1:**
> As noted in the paper, open-world TIR degradation is a non-closed degradation set, where new degradations may continuously emerge.
> Existing all in one TIR restoration methods require joint retraining on historical and new degradations, which increases training cost and requires long-term data storage.
> The motivation for continual TIR degradation restoration is precisely to enable the model to learn only from newly arriving degradations while retaining historical restoration capability.
> Notably, ECMRNet does not forget previously knowledge, because SCGE achieves strict parameter isolation between old and new tasks.
>
> ---
> **W2:**
> Open-world TIR image restoration can be formally defined as a continual degradation restoration problem.
> Let the set of degradations already learned by the model be denoted as $\mathcal{D}=\\{D_1,D_2,\ldots\\}$. A newly arriving degradation is denoted by
> $D_{new}=\\{( X^{new}, Y^{new})\\}$,
> where $X^{new}$ and $ Y^{new}$ represent the degraded images and reference images, respectively. The model is trained only on the current task $D_{new}$, i.e., $f(\cdot \mid \Theta_{new}): X^{new} \rightarrow Y^{new}$.
> The updated model needs to adapt to $D_{new}$, and preserve its restoration capability on all previously tasks.
> We will add this formal definition in the revision.
>
> ---
> **W3:**
> We would like to emphasize that the two settings are not equivalent in terms of the open-world degradation learning problem. Visible restoration benefits from stronger texture redundancy, richer appearance statistics, and natural-image priors, so newly emerging degradations can often still be handled through relatively stable shared representations. In contrast, TIR representations are more sparse and rely more heavily on limited thermal structures, target boundaries, and local thermal contrast; new degradations and their compositions often damage exactly these scarce yet critical thermal cues. Therefore, the key challenge of open-world TIR restoration is not only continual adaptation to new degradations, but also preserving historical restoration ability, controlling long-term model growth, and actively reorganizing historical knowledge for compound degradations. This is precisely the motivation of ECMRNet: SCGE enables interference-free expansion, SEP suppresses ineffective growth, and SKMM mines and recombines transferable sub-degradation knowledge.
>
> ---
> **W4:**
> We argue that ECMRNet is not a simple incremental extension of existing methods. Its novelty lies in four aspects:
> * It targets the underexplored open-world TIR restoration setting rather than closed-set all-in-one restoration;
> * ECMRNet is not merely simple parameter isolation. It adopts a mix-at-ends, separate-in-the-middle architecture that partitions intermediate features into group-wise subspaces, and SCGE expands new groups while freezing historical ones, enabling scalable learning with strict isolation;
> * SEP is not a direct reuse of existing pruning methods. It is tightly coupled with the ECMRNet architecture and introduces structural entropy into redundancy assessment for newly added channel groups. The necessity of each group is defined by the change in 2D SE after removing that group from the current optimal coding structure. To the best of our knowledge, SE has mainly been used in graph information modeling and community detection, and we are not aware of prior work that specifically applies 2D SE to neural parameter or channel pruning;
> * SKMM further enhances compound degradation restoration by selectively mining historical sub-degradation knowledge.
>
> Thus, ECMRNet is a coordinated design for open-world TIR restoration, not a simple combination of parameter isolation and existing pruning techniques.
>
> ---
> **W5:**
> We will redesign Fig. 1 and provide more explicit captions and explanations in the revision.
>
> ---
> **W6:**
> For ECMRNet, knowledge retention is guaranteed by SCGE. SCGE freezes historical channel groups and  introducing newly learnable parameters for new task. Meanwhile, SEP is applied only to the newly added groups. For ECMRNet, the main source of performance deviation is the SEP stage. However, Tab. 3 and Fig. 6 show that it achieves clear parameter/FLOPs reduction with only marginal performance change. Therefore, for ECMRNet, final task performance (Tab. 1) together with pre-/post-pruning deviation is more informative than the forgetting rate.
>
> ---
> **Limitation:**
> We argue that our work's motivation and empirical validation are substantial. This work addresses the underexplored problem of open-world TIR restoration. Experimentally, we construct 7-task incremental degradation sequences on HM-TIR and M3FD, compare against multiple all-in-one and continual learning baselines, validate generalization on three real-world datasets (EN, TIR100, and AWMM), and conduct comprehensive ablations.

---

> > ### Author Rebuttal · Reviewer_BLny · 2026-04-03
> >
> > While I appreciate the authors’ effort to move beyond the standard closed-set restoration setting, I do not think the current paper provides sufficient support for its claims.
> >
> > 1. Although the distinction from conventional closed-set restoration is repeatedly emphasized in both the paper and the responses, neither the benchmark, nor the experimental setting, nor the evaluation addresses key aspects of an open-world setting, such as sequential training, the staged introduction of new degradations, or the resulting forgetting issue. Under the current setup, this work is much closer to open-set restoration under degradation shift than to a broader open-world restoration problem. In my view, the problem formulation is overclaimed.
> >
> > 2. In real-world scenarios, image degradations are typically compositional and continuously varying, rather than cleanly isolated into single degradation type, such as contrast, blur, or noise. The paper, however, evaluates mainly on artificially separated single degradations or a limited set of predefined combinations. It is hard to rigorously ensure that an image contains ONLY contrast degradation, ONLY blur degradation, or ONLY noise degradation. More importantly, even if such controlled settings can be constructed, the space of possible degradation combinations in practice is vastly more complex than the limited cases considered here. As a result, the current experiments can not substantiate the stronger claim of addressing restoration in a open-world setting.
> >
> > I have carefully read the other reviewers’ comments as well as the authors’ response. I found that some the concerns raised by other reviewers are consistent with mine, particularly regarding the fact that the paper does not truly study an open-world setting and that the experiments are insufficient to support such a claim. Therefore, at this stage, I maintain my original score.

---

> > > ### Author Response · Authors · 2026-04-04
> > >
> > > We believe the characterization that the current paper, in terms of its benchmark, experimental setup, and evaluation, does not address key aspects of the open-world setting—such as sequential training, the staged introduction of new degradations, and the resulting forgetting issue—is not accurate.
> > > **First**, we explicitly construct an incremental degradation task sequence, where new tasks are introduced stage by stage rather than being trained and tested once in a static closed-set setting. This is fully consistent with our task definition clarified in **W2**.
> > > **Second**, regarding forgetting, we have already clarified in **W1** and **W6** that ECMRNet achieves strict parameter isolation between old and new tasks through SCGE, thereby preventing forgetting of historical knowledge.
> > > **In terms of evaluation**, we compare not only against multiple all-in-one restoration baselines, but also against four continual learning baselines. In addition, beyond the two synthetic incremental task sequences, we further conduct supplementary evaluations on three real-world degradation datasets, which provide additional evidence of the model’s generalization ability.
> > > **Finally**, we do not believe that the setting studied in this paper can be reduced to mere “degradation shift.” Degradation shift is closer to changes in a known degradation distribution or parameter range, whereas our paper explicitly studies a setting in which different degradation tasks arrive continuously over time, the model is updated continually, historical restoration ability is preserved, and model growth is controlled. These aspects go beyond a simple distribution-shift setting and are precisely the issues targeted by SCGE, SEP, and SKMM in ECMRNet.
> > > **At the same time**, we understand your reminder regarding the boundary of a broader open-world restoration problem. What we validate in this paper is continual adaptation under an expanding set of degradations, rather than direct generalization to arbitrary unseen degradations that have never been learned. We will further clarify this boundary in the revision and narrow the relevant statements to avoid any ambiguity caused by the term “open‑world”.
> > >
> > > ---
> > > We agree that degradations in real-world scenarios are often compositional and continuously varying. Accordingly, **this paper considers not only single degradations but also compound degradations**. The final incremental task sequence consists of three single degradations (C/B/N) and four compound degradations (CB/CN/BN/CBN). We also agree that real images are rarely absolutely pure single-degradation samples. Nevertheless, **controlled single- and compound-degradation settings remain both necessary and standard in restoration research. The existing TIR restoration literature has long studied single degradations (e.g., contrast enhancement, deblurring, denoising) as well as All In One restoration for multiple degradations.** Moreover, we do not rely solely on synthetic settings. We further conduct supplementary evaluations on three real-world compound-degradation datasets. The purpose of these experiments is precisely to provide additional evidence that ECMRNet is robust under real compound degradations, rather than being effective only on “artificially separated pure degradations.” Admittedly, under existing synthetic conditions, we cannot exhaust all degradation combinations that may arise in the real world. However, in our latest response to **Reviewer QuvW**, we further extend the original 7-task sequence by introducing an additional bad-pixel degradation and a more complex compound-degradation task composed of low contrast, blur, noise, and bad pixels, in order to examine the stability of the method under a longer and more challenging task stream. The results show that ECMRNet remains stable on these two newly introduced degradation tasks. Therefore, we believe that the current experiments validate that ECMRNet can achieve continual adaptation over a reproducible and continually expanding degradation set, while also demonstrating strong robustness and generalization on real-world compound-degradation datasets. At the same time, we do not claim that this paper has exhausted or fully covered the entire degradation space under a stronger interpretation of the open-world setting.
> > >
> > > ---
> > > We understand that, after reading the other reviewers’ comments and our responses, you still have reservations about the correspondence between our use of the term “open‑world” and the scope of our experimental support.
> > > However, we believe that characterizing this work as “not truly studying the open-world setting, with experiments insufficient to support such a claim” is still too absolute. In our view, the remaining disagreement is more accurately a matter of calibrating the boundary of the “open-world” wording, rather than whether the paper studies the problem at all.

---

### Official Review · Reviewer_EchJ · 2026-03-25

**Soundness:** 3
**Presentation:** 3
**Significance:** 3
**Originality:** 3
**Overall Recommendation:** 5
**Confidence:** 5

**Summary:**

This paper proposes an Expandable, Compressible, and Mineable Restoration Network (ECMRNet) for open-world thermal infrared (TIR) image restoration. ECMRNet first decomposes intermediate representations into group-isolated subspaces and accommodates new degradations by freezing historical groups while isomorphically expanding new groups, thereby providing sufficient learning capacity for new tasks under strict parameter isolation. It then introduces a novel structural entropy pruning strategy to suppress the unbounded growth of model size as the number of degradation tasks increases. Finally, the paper designs a sub-degradation knowledge mining module to dynamically retrieve and fuse transferable components from historical representations, improving restoration performance under compound degradations. Experimental results on multiple synthetic and real-world datasets show that ECMRNet achieves better restoration performance with fewer parameters and lower computational cost. Overall, the paper is well organized, technically complete, and supported by fairly extensive experiments.

**Compliance With Llm Reviewing Policy:**

Affirmed.

**Final Justification:**

Based on the authors' rebuttal which addressed my primary concerns and reinforced the paper’s technical soundness and originality, I have decided to raise my score to reflect its contributions to open-world TIR image restoration.

**Key Questions For Authors:**

See Weakness.

**Limitations:**

The proposed method is inherently a continual learning framework, meaning that it still needs to be trained on a specific degradation before it can adapt to it. The paper should discuss this limitation more explicitly, as well as whether the proposed framework can be extended to handle unseen degradations beyond those encountered during training.

**Strengths And Weaknesses:**

Strengths:

(1) The paper is well motivated. It addresses an important limitation of existing all-in-one TIR restoration methods: they are not well suited to continuously adapting to newly emerging degradations, while retraining the entire model is computationally and practically expensive. In addition, using a single shared backbone for heterogeneous degradations may introduce task conflict and gradient interference. From the perspective of continual degradation learning, this paper formulates the problem as a self-evolving closed loop of expand–compress–mine, which is both meaningful and well aligned with the open-world setting.

(2) The key components of the proposed method are well connected. SCGE expands channel groups stage by stage in an isomorphic manner, introducing new trainable subspaces for new degradations, while the group-convolution design naturally provides suitable pruning units for SEP. SEP uses two-dimensional structural entropy minimization as a unified criterion to prune channel groups according to their information contribution, enabling controllable and interpretable structural evolution. SKMM further treats historical sub-degradation representations as a retrievable knowledge pool to enhance restoration under compound degradations. The method is coherent and the components are not merely loosely assembled.

(3) The experimental evaluation is relatively comprehensive. The paper reports results on a full task stream including three single degradations and four compound degradations. In addition, it provides experiments and analyses on three real-world degraded datasets. The ablation studies are also reasonably thorough and support the effectiveness and necessity of the main components.

(4) This paper is clearly written and easy to follow. The figures and tables are well designed, and the overall presentation is polished and complete.

Weakness:

(1) The results in Table 1 show that conventional continual learning baselines perform rather poorly. The authors should  supply a clearer discussion of why standard continual learning methods are ineffective for image restoration.

(2) SEP is a key component for controlling model growth. While Figure 6 shows that the pruned model can still converge quickly after pruning, the efficiency and effectiveness of the pruning process itself are not analyzed in sufficient detail.

(3) In the experiments, degradations are learned in a fixed order. However, it remains unclear whether the task learning order affects the final performance. Since this is a continual learning setting, an analysis of order sensitivity would strengthen the empirical study.

---

> ### Author Rebuttal · Authors · 2026-03-29
>
> We sincerely thank you for recognizing the novelty and significance of our work.
> Below we address each of the raised concerns in detail.
>
> ---
> **W1:**
> LwF, EWC, SI, and MAS are all shared-parameter continual learning methods.
> They are often effective for image classification because moderate representation drift may not change the final category prediction.
> In contrast, TIR restoration is a pixel-wise dense prediction problem, where even small feature shifts can noticeably degrade reconstruction quality.
> Moreover, different degradations require inherently different restoration behaviors: denoising suppresses random high-frequency perturbations, whereas deblurring aims to recover corrupted edges and textures.
> Under such heterogeneous objectives, shared-parameter CL methods face an intrinsic trade-off: overly strong constraints hinder adaptation to the new degradation, while weak constraints lead to forgetting of previous degradation knowledge.
> This also aligns with the empirical results in Table 1, where these baselines consistently underperform in the TIR restoration setting.
>
> ---
> **W2:**
> In terms of efficiency, the main overhead of SEP comes from the 2D SE minimization on the group-level similarity graph.
> With a sample pool of N=300, the SEP decision stage takes approximately 21.29 seconds, indicating only modest additional cost.
> In terms of effectiveness, we further compare SEP with magnitude pruning (MP) and sparsity-regularized pruning (SRP).
> For a fair comparison, MP and SRP retain the same number of channel groups as SEP by using a Top-K strategy.
> We evaluate all methods on three single-degradation tasks (Contrast / Blur / Noise) without fine-tuning.
> The results (see table below) show that SEP is overall more effective and more stable.
> In particular, on the blur task, SRP suffers a notable performance drop due to erroneous removal of critical groups.
> We attribute this to SEP’s evaluation of each channel group’s 2D SE contribution to the optimal coding structure, together with its cross-sample frequency-based selection, which leads to more consistent retention decisions.
> Importantly, SEP does not rely on a fixed pruning ratio, but performs adaptive pruning driven by structural information contribution.
>
> **Table 1.**  Comparison of different pruning strategies without fine-tuning.
> | Strategy | training | C-PSNR | C-SSIM | B-PSNR | B-SSIM | N-PSNR | N-SSIM |
> |:---|:---:|:---:|:---:|:---:|:---:|:---:|:---:|
> | MP | × | 19.53 | 0.879 | 19.44 | 0.882 | 17.72 | 0.749 |
> | SRP | √ | 23.05 | 0.784 | 5.36 | 0.053 | 18.85 | 0.709 |
> | SEP (ours) | × | **28.99** | **0.966** | **28.76** | **0.924** | **19.37** | **0.781** |
>
> C/B/N denote Contrast/Blur/Noise, respectively.
>
> ---
> **W3:**
> For single-degradation tasks, ECMRNet is insensitive to the learning order.
> This is because SCGE freezes historical channel groups and expands the network with isomorphic new groups for new degradation learning.
> SEP is also applied only to the newly added channel groups of the current task.
> As a result, the learning order of single degradations does not affect the final learned outcome.
> For compound degradations, SKMM explicitly relies on historical sub-degradation representations as retrievable knowledge.
> Hence its effectiveness presumes that the relevant sub-degradations have already been learned.
> This is not conventional order sensitivity, but rather a structural prerequisite for compound-degradation modeling, which is also consistent with the fact that compound degradations are composed of multiple underlying single degradations.
>
> ---
> **Limitation:**
> In this paper, open-world TIR restoration refers to the ability to continually incorporate and adapt to newly emerging degradations without retraining all previously learned tasks.
> It does not mean that the model can directly restore arbitrary unseen degradations without any training.
> The strength of ECMRNet lies in its ability to accommodate new degradations with controlled model growth while preventing interference with previously acquired knowledge.
> We also agree that direct generalization to completely unseen degradations remains an important direction for future research.
> We will clarify this limitation in the revised manuscript and further discuss possible extensions toward handling unseen degradations.

---

> > ### Author Rebuttal · Reviewer_EchJ · 2026-04-02
> >
> > Thanks to the authors for their responses. The rebuttal has addressed my questions. Overall, the paper is well motivated and presents an effective solution for open-world TIR image restoration. The experimental results support the main claims well. Therefore, I am willing to raise my score.

---

> > > ### Author Response · Authors · 2026-04-03
> > >
> > > We sincerely thank you for your positive assessment of our work and for your willingness to raise the score. We truly appreciate your recognition and will incorporate the relevant clarifications and additional experimental results into the revised manuscript.
> > >
> > > Best regards,
> > >
> > > The authors of Submission 7877

---

### Decision · Program_Chairs · 2026-04-30

**Decision:**

Accept (regular)

**Comment:**

This paper proposes the ECMRNet for open-world thermal infrared (TIR) image restoration. It designs a parameter isolation manner to prevent catastrophic forgetting for all-in-one image restoration methods.

This paper receives mixing scores: 3/4 reviewers recommend a clear/weak acceptance, while 1 reviewer recommends a clear rejection. Overall, the paper is well-motivated and the experiments seem comprehensive and convincing. The main remaining concern (raised by Reviewer BLny) is about the unclear definition and writing quality. In my opinion, although I agree that the clear definition (especially for the first study) is very important for the field, these problems can be clarified in the final version.

Accordingly, I recommend this paper for a weak acceptance. The authors should **clarify the definition and address all concerns raised by Reviewer BLny in the final version**, if the paper is accepted.